# Predicting host range expansion in parasitic mites using a global mammalian-acarine dataset

Pavel B. Klimov [1,2] ✉ & Qixin He [1,2] ✉

Multi-host parasites pose greater health risks to wildlife, livestock, and humans than single-host parasites, yet our understanding of how ecological and biological factors influence a parasite's host range remains limited. Here, we assemble the largest and most complete dataset on permanently parasitic mammalian mites and build a predictive model assessing the probability of single-host parasites to become multi-hosts, while accounting for potentially unobserved host-parasite links and class imbalance. This model identifies statistically significant predictors related to parasites, hosts, climate, and habitat disturbance. The most important predictors include the parasite's contact level with the host immune system and two variables characterizing host phylogenetic similarity and spatial co-distribution. Our model reveals an overrepresentation of mites associated with Rodentia (rodents), Chiroptera (bats), and Carnivora in the multi-host risk group. This highlights both the potential vulnerability of these hosts to parasitic infestations and the risk of serving as reservoirs of parasites for new hosts. In addition, we find independent macroevolutionary evidence that supports our prediction of several single-host species of *Notoedres*, the bat skin parasites, to be in the multi-host risk group, demonstrating the forecasting potential of our model.

Host specificity, quantified by host range breath and the relative susceptibility of hosts, is a fundamental property differentiating single-host (specialist) from multihost (generalist) parasites[1]. Among generalists, the most virulent parasites are those that exhibit equal adaptability to a wide range of hosts, exploiting multiple routes of transmission, and evolving complex patterns of virulence towards each species in their host range[2-4]. The probabilities of emerging infectious diseases are strongly associated with these multi-host pathogens[5]. In extreme cases, host-shifting to an immunologically naïve host species may lead to an epidemic followed by local or global extinction of the new host[6-8]. Remarkably, the source hosts may be largely unaffected in this case because they already have developed natural defenses against the pathogen. Host range also influences parasites' survival strategies on a macroevolutionary time scale. Generally, multi-host parasites tend to persist longer in a community as

diverse host species can maintain these parasites. In contrast, single-host parasites have a higher risk of extinction as their fitness and transmission entirely depend on the host's survival[8-10]. Therefore, they tend to have low/intermediate levels of virulence[11] or evolve into non-virulent commensals or even mutualistic symbionts over evolutionary time[12]. Long-term co-evolutionary interactions between hosts and parasites may favor one of these two strategies, promoting the evolution and maintenance of traits related to infectivity, transmission, and host utilization (in parasites) as well as parasite resistance or avoidance (in their hosts). As a result, parasite lineages may display predictable, trait-related host specificity patterns[13,14] (Fig. 1). In addition, host-parasite associations strongly depend on host phylogenetic, geographic, and environmental factors[15-17]. Pathogen sharing, for example, could be strongly influenced by pairwise host phylogenetic similarity and spatial co-occurrence[18]. Thus, to predict a parasite host

[1]Lilly Hall of Life Sciences, Purdue University, 915 Mitch Daniels Blvd, West Lafayette, Indiana 47907, USA. [2]These authors contributed equally: Pavel B. Klimov, Qixin He. ✉e-mail: pklimov@purdue.edu; heqixin@purdue.edu

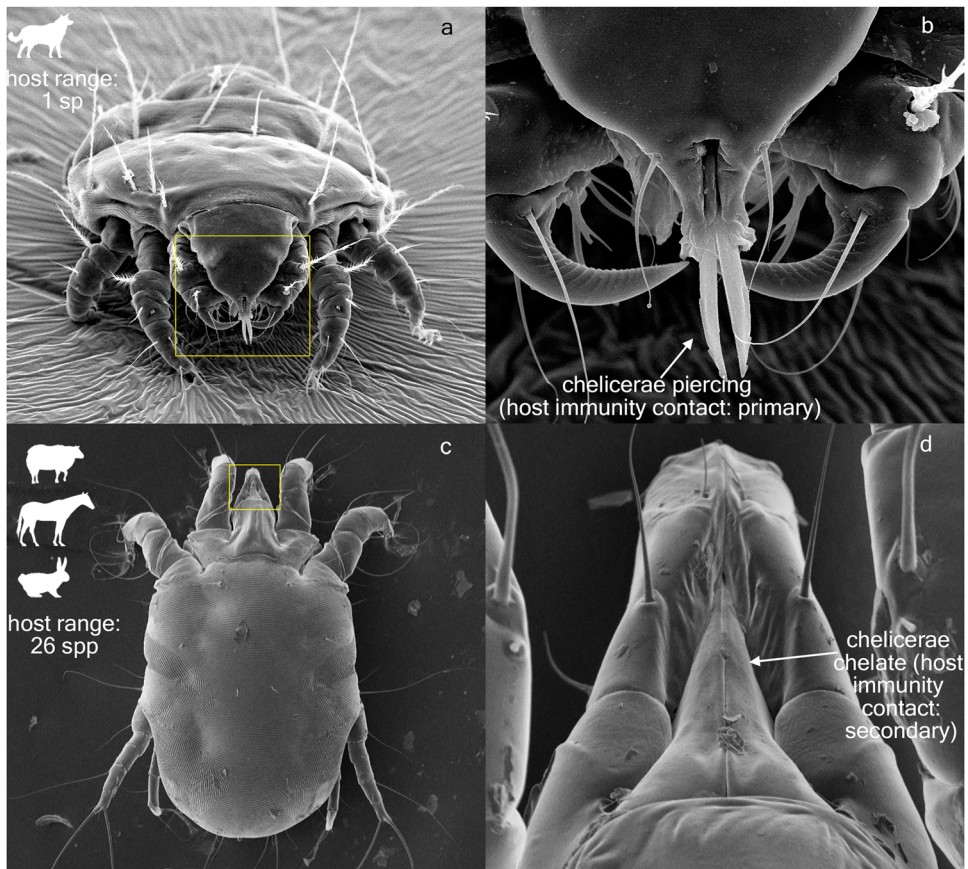

**Fig. 1 | Characteristics affecting mite host range. a** single-host parasite, *Cheyletiella yasguri*, parasitizing the dog *Canis familiaris*. Yellow square is enlarged in **b**, mouthparts of *Cheyletiella yasguri*, showing piercing chelicerae (primary host immunity contact). **c**, multi-host parasite, *Psoroptes ovis*, parasitizing 26 host species. Yellow square is enlarged in **d**, mouthparts of *Psoroptes ovis*, showing chelate chelicerae (secondary host immunity contact).

range, variables related to the pathogen, host, and environment should be considered simultaneously in order to build a robust probabilistic model[14].

Accurate prediction of a parasite's host range can contribute to our understanding of which parasites are capable of a host shift and causing new diseases. This knowledge may translate into better forecasting infectious disease emergence and preventing parasite spillover, especially to humans and domestic animals[19]. Recent research that used a probabilistic modeling framework has identified several predictors of the host-parasite links, mainly focusing on mammalian viruses[14,18–27]. Despite progress extolled by these advanced modeling approaches, it still remains unclear whether host range predictions drawn from these studies are applicable to other host-pathogen systems. There are also two major but commonly overlooked technical challenges related to host-parasite data analyses: imbalanced classification and unobserved multi-host parasites. In imbalanced classification, one class is more prevalent than the other, potentially leading to a bias in model building and a deceptively high model accuracy. Despite the prevalence of imbalanced datasets, only a single study has explicitly accounted for this aspect so far[28]. More importantly, in host-parasite systems, not all host and infectious agent species are known, leading to the concept of 'epidemiological dark matter'[29]. This lack of knowledge, particularly when true multi-host parasites are erroneously classified as single-hosts due to insufficient sampling (referred to as 'unobserved multi-hosts'), can significantly impact the predictive power of analyses modeling host range expansion, host specificity, or host sharing. However, this issue has received little attention thus far, as most models assume that both single-and multihost classes are correctly labeled, but see Dallas et al. [30].

Here, we assemble a large dataset of all known acariform mite-mammal associations (1,984 mite and 1,432 mammal species) and conduct host range modeling in a global, pan-host, and pan-parasite context, which means all known target parasites and all their known hosts are included (unlike many mammal-virus analyses[14], where avian hosts have been omitted by design). We focus only on permanent mammal associates−full-time parasites, staying on the host for the entire life-cycle, lacking a free-living or dispersal stage, and colonizing host individuals on contact (Fig. 1). Our dataset is one of the largest and most complete among compatible host-parasite databases available to date; for example, it has 92 primate-associated target mite species, while other databases reported much fewer species: 34[31] or one species[32]. Among them is *Sarcoptes scabiei*, one of the most virulent multi-host mites, responsible for epidemics in wild and domesticated mammals and skin disease outbreaks in humans[33].

We populate this dataset with a set of predictor variables related to parasites, their hosts, and the environment, which can aid in predicting mites' host range expansion. For example, among these predictors, two variables are related to mites' feeding specialization − mites with direct immune system interactions (e.g., hair follicular mites) are expected to have a lower establishment probability (and therefore narrower host range) compared to those feeding on non-immunogenic host tissue derivatives (e.g., fur mites). Similarly, ectoparasitic mites with diverse dispersal stages and broad geographic distributions are expected to have higher probabilities for successful transmission to new hosts, potentially contributing to broader host ranges. Hosts with many closely related mammal species provide more opportunities for their parasitic mites to invade additional host species due to similar immune evasion mechanisms[25]. Likewise, hosts living in

regions with a high concentration of sympatric mammal species present direct opportunities for host shifting of their parasites, leading to broader host ranges. Other host properties, such as litter size, domestication status, and living in anthropogenically disturbed areas also offer further avenues for mite transfers and host range expansions. Finally, abiotic factors, such as temperature and humidity, may affect mite survival outside the host during transmission, thereby facilitating or prohibiting the transmission process, leading to broader or narrower host ranges, respectively.

To address the imbalanced nature of most host-parasite associations, we extend the previous modeling efforts[14,18] by using down- and up-sampling strategies and a metric not influenced by single vs multi-host class imbalance. We also employ several strategies to alleviate the potential effect of unobserved multi-hosts: positive-unlabeled (PU) learning (explicitly assumes that labels in the multi-host class are known, while the other class is unlabeled and may contain both single-host and multi-host instances), weighting by publication counts (downweighs mite species with less sampling efforts as estimated by relevant publication numbers), and down- and up-sampling (maximizing the prediction of multi-hosts, which is the class of interest).

Based on a set of 13 predictor variables related to parasites, their hosts, and the environment, and accounting for the above challenges, such as class imbalance and potential influence of unobserved multi-hosts, we build a statistical model to predict the likelihood that a single-host parasite can extend its host range to become multi-host. To evaluate the predictive power of our best model, we conduct simultaneous k-fold cross-validation using an independent and unmodified test (holdout) dataset. Because our best model was based on resampling, using the unmodified test dataset for model evaluation should provide performance estimates that better reflect real-world situations. We then use our predictive model for forecasting to identify (i) a set of observed single-host species that have a high probability of becoming multi-host (high host switching / host range expansion risk), and (ii) a set of mammalian lineages that are 'enriched' with risk group mites. We then discuss whether our forecasting results, which identify potentially emerging epidemic risk-group parasites, could be supported by independent lines of evidence, such as host ecology promoting extensive parasite exchange, and whether related lineages have experienced host shifts on a macroevolutionary scale.

## Results

### Model selection

We evaluated the overall performance of five models using five-fold cross-validation and then assessed their predictive power with an independent test dataset (Table 1). A logistic regression analysis using generalized linear model served as the baseline (Model Id=1 in Table 1), while the four other models employed various strategies to address the issues related to class imbalance and potential unobserved multi-hosts: (2) weighting by publication counts to down-weight less-sampled and studied mite species, employing either down- (3) or up-sampling (4) during each iteration of the cross-validation procedure to enhance the prediction accuracy of the minority class (*multihost*), and (5) using a positive-unlabeled learning model with AdaSampling and an SVM classifier, assuming that only the positive class (*multihost*) is labeled, while the negative class is an unlabeled mixture of true single-hosts and unobserved multi-hosts.

The baseline model (Model Id=1 in Table 1) exhibited deceptively higher overall accuracy (0.810), which was influenced primarily by the majority class (*singlehost*), as indicated by high specificity (0.940) and low sensitivity (0.475). The two resampling procedures, up and down-sampling (Model Ids 3&4 in Table 1), largely removed this imbalance, resulting in sensitivities of 0.664-0.680 and specificities of 0.773–0.779, thus significantly improving the prediction of the category of interest, *multihost*. The PU model (Model Id=5 in Table 1) optimized the prediction of the *multihost* class, achieving the highest

**Table 1 | Five-fold cross-validation of five models using no-sampling, weighting, down- and up-sampling, and a positive-unlabeled (PU) approach**

| Id | Resampling | Train:Singlehost | Train:Multihost | AUC (CI) | Accuracy | Sensitivity | Specificity | F1 |
|---|---|---|---|---|---|---|---|---|
| 1 | none | 733 | 291 | 0.7769 (0.7240-0.7769) | **0.8052** | 0.4754 | **0.9398** | 0.5859 |
| 2 | none(weighted) | 733 | 291 | 0.7735 (0.7208-0.7735) | 0.7933 | 0.5082 | 0.9097 | 0.5877 |
| 3 | down* | 291 | 291 | **0.7986 (0.7493-0.7986)** | 0.7506 | 0.6803 | 0.7793 | **0.6125** |
| 4 | up | 733 | 733 | 0.7792 (0.7265-0.7792) | 0.7411 | 0.6639 | 0.7726 | 0.5978 |
| 5 | PU ada | 733 | 291 | 0.7882 (0.7384-0.7882) | 0.7387 | **0.7049** | 0.7525 | 0.6099 |

The initial train dataset has 1,024 records, while the test (holdout) dataset has 421 records; the positive class is *multihost* (minority class); Google Scholar publication counts were used as weights in the weighted model. Best values are emphasized in bold.
*=preferred model.

sensitivity (0.705) but the lowest specificity (0.753) among the resampling and weighting-based models (Model Ids 5 vs. 2–4 in Table 1). Based on the Area Under the ROC Curve (AUC) and F1 score, the down-sampled model (Model Id = 3 in Table 1) performed best, with an AUC of 0.799 vs 0.774–0.788 and an F1 score of 0.613 vs 0.586-0.610 (Table 1; Fig. 2a), indicating that this model has the best overall prediction for both *multihost* and *singlehost* classes. Therefore, we selected the down-sampled model as our preferred model. Alternatively, we also compared its forecasting results with those of the PU model, which maximized the accurate prediction of the *multihost* class.

### The preferred model

The likelihood ratio test suggests that our preferred model is significantly different from the intercept-only model ($p < 2.2e-16$), indicating that the independent variables, as a whole, contribute significantly to the prediction of the mite host range. Lower Akaike's Information Criterion (AIC) and residual deviance values vs. those for the intercept-only model also indicate an overall good model fit (AIC 601.71 vs 808.8, residual deviance 563.71 vs. 806.8). Standard errors for the coefficient estimates were substantially less than 2.0, except for the spline variable ns(Average Temperature, 2)1 (Table 2), indicating the absence of a high multicollinearity. Three categorical and four continuous predictors were significant, one continuous predictor was marginally significant, while other variables were nonsignificant, including *Domesticated Host* (Table 2; Fig. 2b).

### Factors affecting the odds of becoming multi-host

Individual model coefficients for mite-related predictors were positive for all levels of mite-related independent variables, *Host Immunity Contact Level* (*primary* vs *secondary* or *none*), *Parasitism* (*ectoparasitic* vs *endoparasitic*), *Mite Biogeographic Region* (*multiregion* vs *singleregion*), *Mite Dispersal Stage* (*immatures* vs *female* or *female+immatures*), *Precopulatory Guarding* (*absent* vs *present*) (Ids 1-5 in Table 2, Fig. 2b, Supplementary Fig. 1 a-e), indicating their positive effect on becoming multihost. For example, all else being equal, the odds of a mite with *secondary* host immune system contact being *multihost* are Exp (2.03) = 7.60 times higher than those with *primary* contact (where 2.03 is the model coefficient estimate, Id=1.1 in Table 2). All model coefficients were statistically significant ($p < 0.05$), except for *Mite Dispersal Stage* and *Precopulatory Guarding* (Table 2). Thus, having *secondary* or *none* host immunity contact level (or having chelate chelicerae), being an ectoparasite, and having a multi-region geographic distribution all increase the odds of a single-host mite becoming multi-host (Fig. 2 c-e). These results agree with our a priori expectations (see the Material and Methods section).

Of the five host-related predictors (Ids 6-10 in Table 2, Supplementary Fig. 1 f–j), only two, *Co-Distributed Potential Hosts, Phylogenetically Similar Potential Hosts*, were statistically significant (Ids 6&7 in Table 2, Fig. 2f, g), while the second natural spline basis function of *Host Body Mass log* was marginally significant (Id=10.2 in Table 2, Fig. 2h). This suggests that a mite is statistically more likely to become multi-host if its observed host(s) (i) have many geographic overlaps with other potential hosts and (ii) are phylogenetically similar to many other potential hosts on the mammalian tree of life, regardless of whether these hosts co-occur. Each additional sympatric potential host increases the odds of becoming multi-host by 3.6%, and each additional phylogenetically similar potential host increases the chance of becoming multi-host by 86% (Ids 6&7 in Table 2). However, other host-related variables (or natural spline functions) had non-significant coefficients, therefore, their effect on the outcome variable is uncertain (Ids 8, 9 &10.1 in Table 2, Supplementary Fig. 1 h-j).

Two climatic variables, *Average Precipitation* and *Average Temperature*, were natural spline (df=2) variables (Ids 11-12 in Table 2, Fig. 2i, Supplementary Fig. 1 k, l). Briefly, a natural spline function, is a

flexible curve-fitting technique that smooths the data without imposing strict assumptions about the relationship between the predictor variable and the outcome variable. This approach is especially useful when the relationship is nonlinear or when the effect of the predictor can change directionality. For example, in scenarios where midrange temperatures are optimal and have a positive effect on the outcome, while both low and high temperatures diminish this effect, a natural spline can effectively capture this nonlinear relationship. The marginal effect of the variable *Average Temperature* (mean monthly temperature x0.1 °C) showed this pattern−it was fit by a hump-shaped curve with an upturn (scarce data), a peak around 11.8 °C (117.5*0.1), and a downturn (dense data) (Fig. 2i). In other words, in areas with average temperatures above 11.8 °C, the odds of becoming multi-host decrease, while in areas having lower temperatures, the effect of this variable is negative, which agrees with our a priori expectation. The *Average Precipitation* variable (mean monthly precipitation, mm) was fitted as a shallow, concave downward arch with a peak at 173 mm/month (Supplementary Fig. 1 k). This contrasts with our a priori expectation of a linear, positively correlated relationship with the outcome. However, this predictor was not statistically significant (Id=11 in Table 2).

The log-transformed, *Average Human Population Density log* (a proxy for anthropogenic habitat disturbance) had a significant positive linear effect on the dependent variable (Fig. 2 j), which was expected a priori.

### Predictions from the model: unobserved multi-host mites

When a mite species is labeled as *singlehost* in our database, it is still possible that it has other unobserved host species due to varying sampling efforts. To detect potentially unobserved multi-hosts (a risk group), we combined our prediction and test (holdout) datasets and calculated counts of correct and incorrect classifications for each class of the dependent variable (Fig. 3a, b). Unobserved multi-hosts were entered into the analysis in the *singlehost* category but classified as *multihost* by the model along with truly misclassified single-hosts (Fig. 3a, b). There were 220 counts of *singlehost* classified as *multihost* and 120 counts in the category of *multihost* classified as *singlehost* (Fig. 3b). To be conservative in estimating potential host shifts, out of 220 counts of *singlehost* classified as *multihost*, we only included those with a model prediction probability above 0.7 resulting in a subset of 86 instances included in the multi-host risk group. We then analyzed whether certain host orders are 'enriched' in this group. Per-host-order risk analyses identified Rodentia, Chiroptera and Carnivora as host orders with disproportionately high numbers of unobserved multi-hosts, while marsupials (Diprotodontia) lack any single-host mite species predicted in the multi-host risk group (Fig. 3c, Table 3). Among these mites, were five sarcoptid skin mites parasitizing molossid or vespertilionid bats: *Notoedres ovatus* (host *Mops condylurus*), *N. yunkeri* (*Molossus molossus*), *N. anisothrix* (*M. molossus*), *N. helicothrix* (*Cynomops planirostris*), and *Notoedres eptesicus* (*Eptesicus brasiliensis*). These mites are currently cited in the literature as single-host parasites, but our preferred model predicted them to be multi-host with high probabilities, ranging from 0.702 to 0.932 (Fig. 3b, Supplementary Data 1). Our auxiliary PU learning model, which assumes that only the *multihost* class is correctly labeled, predicted these five species in the multi-host risk group with even higher probabilities, 0.872-0.994 (Supplementary data 2).

### Mite-sharing patterns among hosts

We further investigated how patterns of mite sharing among mammalian hosts are shaped by the host pairwise geographical overlaps and phylogenetic distances (PD). The mite-host association network was projected to build a unipartite host-host network from our database, in which an edge represents a host pair that shares one or more mite species. Logistic regression using a generalized linear model was

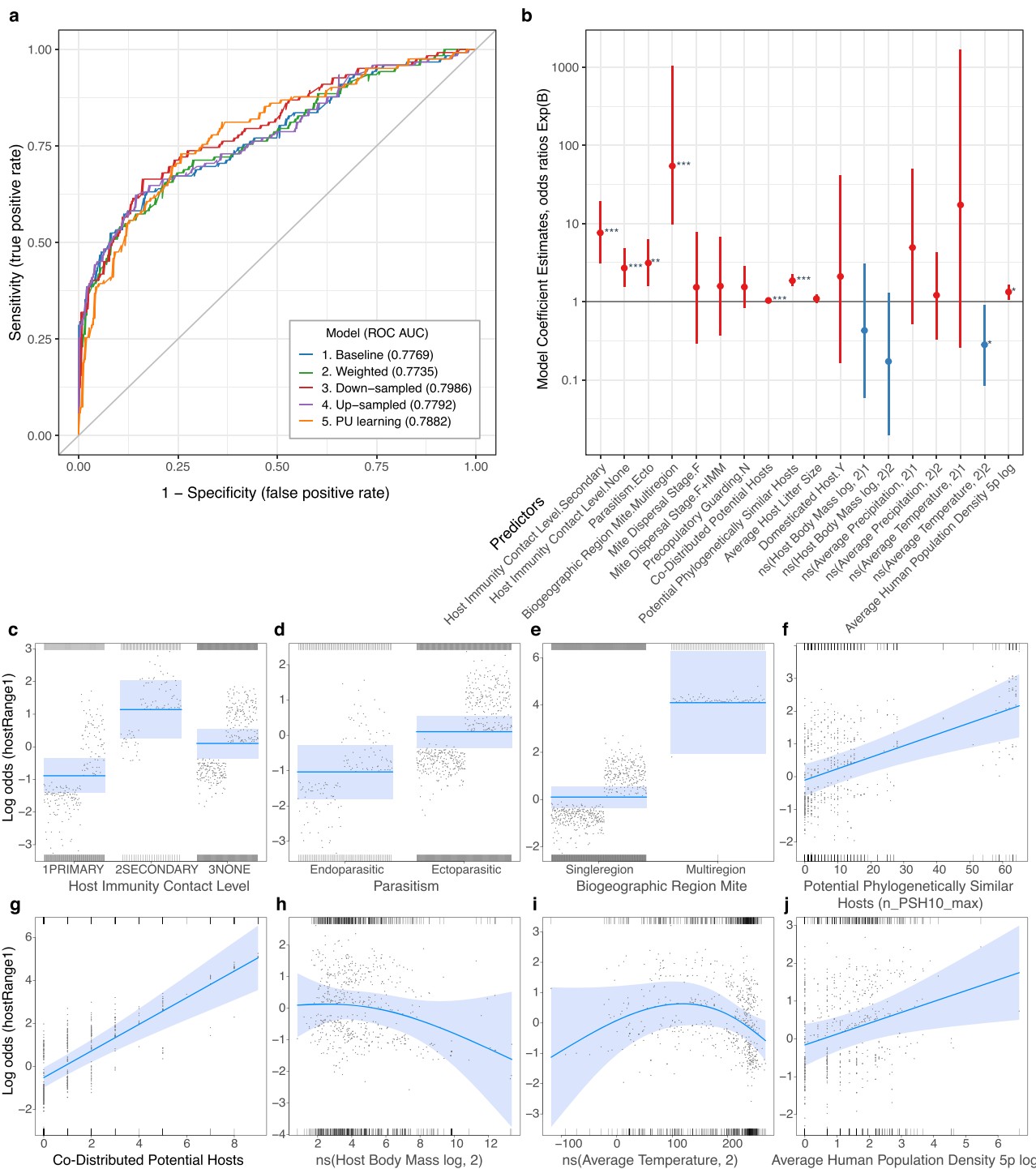

**Fig. 2 | Performance of models predicting the probability of becoming multihost and the preferred model. a**, Performance of five models (no-sampling, weighting, down-, up-sampling, PU learning) estimated using a Receiver Operating Characteristic (ROC) analysis; the down-sampled model has the largest ROC area under curve (AUC) and is considered as our preferred model. Detailed model statistics are given in Table 1. **b** Exponentiated coefficient estimates (dots) and their standard errors (bars) of the preferred model; positive (red) or negative (blue) effects of the 13 factors on predicting the *multihost* category; data are given as odds ratios; bars show standard errors; bars crossing the x-axis at zero are not statistically significant; two-sided z-score statistical significance is indicated as follows: *** $p < 0.001$; ** $p < 0.01$; * $p < 0.05$. **c–j** Relationships between the outcome (*multihost*) and a set of predictors based on the preferred model; categorical predictors are shown as bars, continuous predictors are shown as lines; predictor variables may increase or decrease the odds of becoming multi-host, for example, the variable *Co-Distributed Potential Hosts* (g) increases the odds of becoming multi-host at its whole range, while the natural spline variable *Average Temperature* (mean monthly temperature x0.1 °C) increases (range from −118.3 to +117.5) or decreases (range from +117.5 to +269.6) the odds of becoming multi-host; the effect of each variable is shown on a scale of odds ratios as the remaining explanatory variables are held constant (blue lines); 95% confidence bands are shown as shaded blue; partial residuals (the sum of the residuals and predictor terms) are shown as dots; bottom and top rugs correspond to the observations of class 0 residuals (*singlehost*) or 1 (*multihost*), respectively; ns(var, 2) denotes a two-degree-of-freedom (df=2) natural spline. Source data are provided as a Source Data file.

**Table 2 | Preferred model predicting the probability of a single-host mite to become multi-host using a set of mite-related, host-related, and environmental variables and a majority class down-sampling strategy (Table 1: Model Id=3)**

| Id | Coefficients | Estimate | CI.LL | CI.LB | SE | Exp (B) | z value | Pr(>|z|) | Sign |
|---|---|---|---|---|---|---|---|---|---|
| | (Intercept) | −6.157 | −9.161 | −3.339 | 1.483 | 0.002 | −4.152 | 3.29E-5 | *** |
| 1.1 | Host Immunity Contact Level.Secondary | 2.028 | 1.130 | 2.962 | 0.466 | 7.600 | 4.352 | 1.35E-5 | *** |
| 1.2 | Host Immunity Contact Level.None | 0.990 | 0.431 | 1.564 | 0.288 | 2.692 | 3.434 | 0.0006 | *** |
| 2 | Parasitism.Ecto | 1.139 | 0.469 | 1.838 | 0.348 | 3.125 | 3.272 | 0.0011 | ** |
| 3 | Mite Biogeographic Region.Multiregion | 3.996 | 2.293 | 6.953 | 1.085 | 54.363 | 3.683 | 0.0002 | *** |
| 4.1 | Mite Dispersal Stage.F | 0.426 | −1.222 | 2.063 | 0.832 | 1.530 | 0.511 | 0.6091 | |
| 4.2 | Mite Dispersal Stage.F_IMM | 0.458 | −0.995 | 1.913 | 0.734 | 1.581 | 0.624 | 0.5325 | |
| 5.2 | Precopulatory Guarding.N | 0.432 | −0.178 | 1.049 | 0.312 | 1.540 | 1.383 | 0.1666 | |
| 6 | Co-Distributed Potential Hosts | 0.035 | 0.019 | 0.051 | 0.008 | 1.036 | 4.259 | 2.06E-5 | *** |
| 7 | Phylogenetically Similar Potential hosts | 0.619 | 0.458 | 0.798 | 0.087 | 1.856 | 7.140 | 9.36E-13 | *** |
| 8 | Average Host Litter Size | 0.089 | −0.032 | 0.210 | 0.061 | 1.093 | 1.447 | 0.1479 | |
| 9 | Domesticated Host.Y | 0.742 | −1.804 | 3.726 | 1.392 | 2.100 | 0.533 | 0.5939 | |
| 10.1 | ns(Host Body Mass log, 2)1 | −0.849 | −2.826 | 1.125 | 1.005 | 0.428 | −0.844 | 0.3984 | |
| 10.2 | ns(Host Body Mass log, 2)2 | −1.761 | −3.945 | 0.250 | 1.063 | 0.172 | −1.657 | 0.0976 | . |
| 11.1 | ns(Average Precipitation, 2)1 | 1.594 | −0.654 | 3.927 | 1.166 | 4.925 | 1.368 | 0.1713 | |
| 11.2 | ns(Average Precipitation, 2)2 | 0.190 | −1.099 | 1.460 | 0.649 | 1.209 | 0.293 | 0.7696 | |
| 12.1 | ns(Average Temperature, 2)1 | 2.849 | −1.345 | 7.433 | 2.240 | 17.273 | 1.272 | 0.2035 | |
| 12.2 | ns(Average Temperature, 2)2 | −1.268 | −2.466 | −0.099 | 0.602 | 0.281 | −2.104 | 0.0354 | * |
| 13 | Average Human Population Density log | 0.287 | 0.062 | 0.513 | 0.114 | 1.332 | 2.506 | 0.0122 | * |

The significance of estimated coefficients is indicated using z-score tests. Reference categories: *Host Immunity Contact Level.Primary* vs *Secondary* and *None*, *Mite Dispersal Stage.IMM* vs *F* and *F_IMM*, *Parasitism.Endo* vs *Ecto*, *Precopulatory Guarding.Y* vs *N*, *BiogeographicRegionMite.Singleregion* vs *Multiregion*, *Domesticated Host.N* vs *Y*. log=natural logarithm transformation; ns(var,2) =natural spline with two degrees of freedom; CI = confidence interval, lower (LL) and upper limit (UL); SE = standard error; Exp(B) = odds ratios for the predictors (exponentiated model coefficients); z value = ratio of the estimated coefficient to its standard error (absolute value represents the variable importance in the model); Pr = probability of observing a z value (two-sided) as extreme or more extreme than the observed value assuming that the coefficient is zero (null hypothesis); a *p*-value less than 0.05 indicates that the null hypothesis can be rejected; significance (sign.) codes: ***p < 0.001; **p < 0.01; *p < 0.05; p < 0.1.

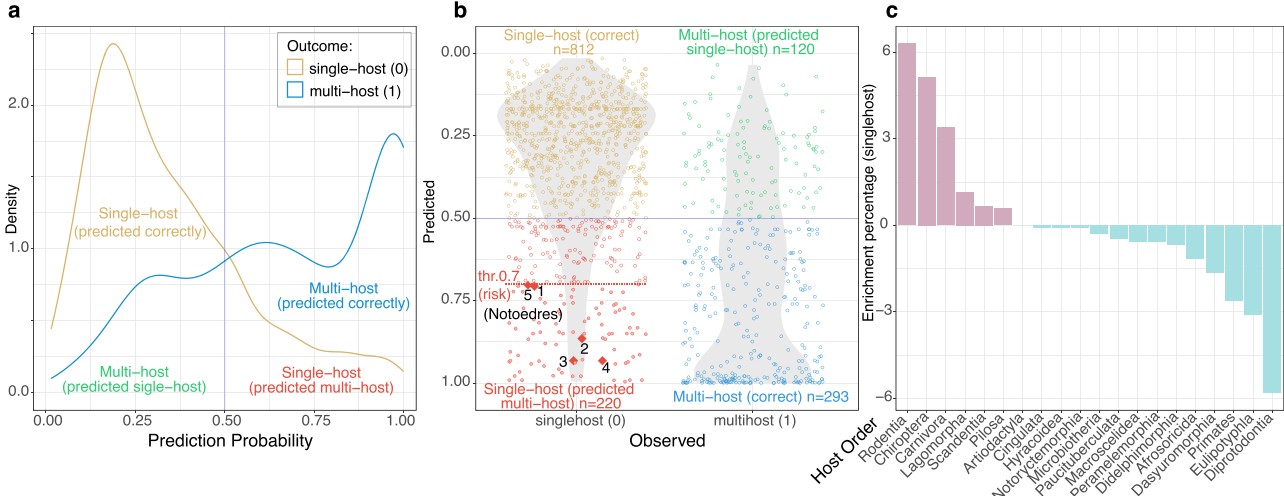

**Fig. 3 | The accuracy of predicting the *singlehost* and *multihost* classes of the outcome variable, and high epidemic risk-group mites per host order based on the preferred model. a** Distribution of prediction score grouped by known outcome, *singlehost* (0) and *multihost* (1); **b** Confusion matrix with cutoff at 0.5, high epidemic risk-group mites (potential multi-hosts, probability threshold at 0.7) and potential risk-epidemic parasites (*Notoedres*) are emphasized; 1=*Notoedres helicothrix*, 2 = *N. ovatus*, 3 = *N. yunkeri*, 4 = *N. anisothrix*; 5 = *N. eptesicus*. Grey shadings represent violin plots of the distributions of predicted likelihood of *multihost* of all the mite species. **c**, High epidemic risk-group mites per host order. Positive values (percentages) indicate higher than average likelihood of an observed single-host actually being a multi-host parasite or having the potential of being multi-host. The classification probability threshold was set to 0.7. Source data are provided as a Source Data file.

then used to predict the probability of mite-sharing between host pairs given their geographic overlap and phylogenetic distances (see Material and Methods for details). Only hosts that share at least one mite species with other hosts were kept for the mite-sharing analyses. Overall, the probability of mite-sharing increases with smaller phylogenetic distances and larger geographic overlap significantly (Fig. 4; Supplementary Table 1), with a negative interaction between the two predictors. We found that phylogenetic distance between host pairs is the most important factor in determining mite-sharing probabilities (Fig. 4a). Regardless of the degree of geographic overlap, the sharing probability quickly drops to zero when phylogenetic distance is larger than 100 (this threshold roughly corresponds to be within the same order, Fig. 4b). Alternatively, allopatric but highly phylogenetically similar (PD < 10) host pairs still have more than a 50% chance of sharing mites (Fig. 4b). The sharing probability increases to over 0.75 when the host pair is fully sympatric, i.e., geographic overlap = 1.

## Discussion

Much of recent research has been focused on predicting the likelihood of emerging infections using host specificity or host range expansion as one of the risk factors, directly relating host shifting abilities and infection of novel hosts[5,14,34]. A variety of parasite, host, ecological and evolutionary determinants have been suggested as potential predictors of host range of parasitic organisms: host immunity[20], species abundance[21,22], body size[23], sex[24], geographic range, sympatry and host phylogenetic similarity[14,18,25], parasite transmission strategy[26], various environmental conditions[27], and human population size within a parasite species geographical range[18]. Incorporating these factors to describe the level of host specificity in a quantitative modeling framework can be beneficial for forecasting infectious disease emergence, parasite spillover events, and can also help in preventing or reducing transmission of multi-host pathogens to humans and domestic and wild animals[19].

Although forecasting host range expansion in single-host parasites is important from an epidemiological perspective, this task is challenging even with the use of advanced probabilistic modeling approaches. First, new epidemics are rare events despite many potentially dangerous parasites identified by the model. An epidemic depends on many factors, including the ability of the host immune system to effectively defend the host against a new pathogen, the ability of the parasite to evade the immune surveillance of a novel host, the host population size and contact frequency. An accurate representation of these predictors involves a detailed understanding of host-parasite biology. Second, as parasites may have hidden potential to infect novel hosts, a model trained on the current associations may not have sufficient predictive power when applied to future unobserved events. In addition, single-host parasites may be unobserved multi-host parasites, which could introduce noise during model building, resulting in diminishing accuracy in predicting the multi-host

## Table 3 | Potentially unobserved multi-hosts (single-hosts predicted as multi-hosts) at four probability thresholds (Thr 0.5-0.9) per host order

| Host Order | Single-Host | ΔThr0.5 | ΔThr0.7 | ΔThr0.8 | ΔThr0.9 |
|---|---|---|---|---|---|
| Rodentia | 39.05 | 2.31 | 6.30 | 10.95 | 14.28 |
| Chiroptera | 32.07 | 5.65 | 5.14 | 2.41 | −5.41 |
| Carnivora | 2.42 | 1.21 | 3.39 | 6.20 | 14.24 |
| Lagomorpha | 1.16 | −0.25 | 1.16 | 0.56 | 2.17 |
| Scandentia | 0.48 | −0.03 | 0.68 | −0.48 | −0.48 |
| Pilosa | 0.58 | 0.33 | 0.58 | −0.58 | −0.58 |
| Artiodactyla | 1.16 | −0.71 | 0.00 | 0.56 | −1.16 |
| Cingulata | 0.10 | −0.10 | −0.10 | −0.10 | −0.10 |
| Hyracoidea | 0.10 | −0.10 | −0.10 | −0.10 | −0.10 |
| Notoryctemorphia | 0.10 | −0.10 | −0.10 | −0.10 | −0.10 |
| Microbiotheria | 0.29 | 0.16 | −0.29 | −0.29 | −0.29 |
| Paucituberculata | 0.48 | 0.42 | −0.48 | −0.48 | −0.48 |
| Macroscelidea | 0.58 | −0.58 | −0.58 | −0.58 | −0.58 |
| Peramelemorphia | 0.58 | −0.58 | −0.58 | −0.58 | −0.58 |
| Didelphimorphia | 1.84 | 0.43 | −0.68 | −0.12 | −1.84 |
| Afrosoricida | 1.16 | 0.66 | −1.16 | −1.16 | −1.16 |
| Dasyuromorphia | 1.65 | −1.65 | −1.65 | −1.65 | −1.65 |
| Primates | 4.94 | −1.31 | −2.62 | −4.94 | −4.94 |
| Eulipotyphla | 5.43 | 0.03 | −3.10 | −3.70 | −5.43 |
| Diprotodontia | 5.81 | −5.81 | −5.81 | −5.81 | −5.81 |
| **Observations (n)** | 1032 | 220 | 86 | 58 | 30 |

Threshold values are given as differences (Δ) between the percentage of single-hosts identified as multi-host at a particular threshold per host order and the percentage of observed single-hosts per host order (column: Single-Host). Positive threshold values indicate higher than average likelihood of an observed single-host actually being a multi-host parasite in a given source host order. All values are given as percentages, except for Observations, which are counts. Data at threshold 0.7 are visualized in Fig. 3b.

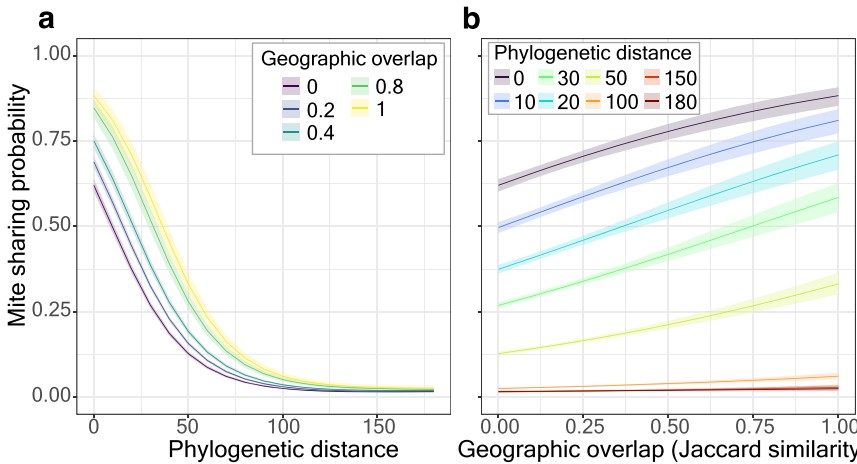

**Fig. 4 | Mite sharing probability and pairwise host characteristics, phylogenetic distance and geographic overlap. a** Mite sharing probability (colored lines) decreases with host phylogenetic distance at five levels of host geographic overlap. **b** Mite sharing probability (colored lines) increases with host geographic overlap at eight levels of host phylogenetic distance. The shaded bands are standard errors of predicted sharing probability. Source data are provided as a Source Data file.

class. Third, failure to account for the imbalanced observations of single- and multi-host pathogens may lead to models with classification accuracy biased toward the majority class (i.e., single-host pathogens).

Our model emphasizes an accurate prediction of the risk group (multi-host) by using down-sampling, a technique that equalizes both single- and multi-host categories by randomly removing observations from the majority class (single-host), thus potentially reducing the noise (unobserved multi-hosts classified as single-hosts). This technique also accounts for class imbalance which is frequently present in host-parasite datasets. Our down-sampling approach was found to be superior to other noise-reducing approaches, such as positive-unlabeled (PU) learning, up-sampling or weighting records by previous research effort (Table 1: AUC, F1). As estimated by independent holdout validation, for the down-sampled model (Model Id=3 in Table 1), metrics related to multi-host class prediction (sensitivity=0.680) was the highest among all other models (except for the PU learning model), while still having a reasonable accuracy in predicting the single-host class (specificity=0.779). In contrast, the baseline model (no noise reduction) classified the minority class (*multihost*) worse than a random guess (Model Id=1 in Table 1; sensitivity=0.475), while having a deceptively high overall accuracy (0.805), which was driven by the majority class (Model Id=1 in Table 1; specificity=0.940).

By linking various predictors describing the parasite morphology, distribution, host and parasite ecology, the number of phylogenetically similar hosts and co-distributed hosts, as well as climatic and anthropogenic habitat disturbance, our model provides a unique window into the determinants affecting the ability of a single-host parasite for host shifting and thus becoming multi-host. In order of importance (Table 2b: absolute values of column 'z-value'), the likelihood for becoming multi-host increases in mites: (i) having a large number of potential phylogenetically similar hosts (Id=7 in Table 2); (ii) feeding with chelate chelicerae on host tissues having no direct contact to the host immune system but an immune response still can be mounted if the tissue is damaged (e.g., outermost layer of the epidermis consisting of dead keratinocytes) (Id=1.1 in Table 2); (iii) having a large number of co-distributed potential hosts (Id=6 in Table 2); (iv) spatially distributed in more than one biogeographic region (Id=3 in Table 2); (v) inhabiting host tissue derivatives which have no contact to the host immunity (e.g., fur and other hair derivatives) (Id=1.2 in Table 2); (vi) being an ectoparasite (Id=2 in Table 2); (vii) living in areas with a larger anthropogenic habitat destruction/larger human population density (Id=13 in Table 2); (viii) living in warm rather than hot regions (applies only to areas with an average monthly temperature of 21.8 °C and above) (Id=12.2 in Table 2); (ix) associated with less massive hosts (marginally significant, applies only to hosts with an average mass of 53.3 g and above) (Id=10.2 in Table 2). In this list, the first most important variable, *Potential Phylogenetically Similar Host*s, characterizes phylogenetic affinities of potential hosts. The second most important predictor (ii) is related to both mite proximity to the host immune system and mite mouthpart morphology, the two mite properties that are nearly perfectly correlated to each other. Similarly, the majority of statistically significant predictors identified by our model were related to mite (ii, iv-vi) or host properties (i, iii, ix), while only a single predictor was related to each climate (viii) and habitat disturbance (vii). In contrast to the literature, which suggests that host domestication status[10,11,26] and host body mass[14] or size[23] are important risk factors of host range expansion, our study found that these two predictors are not statistically significant (Ids 9&10 in Table 2). In case of host domestication, the effect of this variable on the outcome was positive (as expected), however, as most host shifts onto domesticated animals have been made by multi-host parasites, the model coefficient was estimated non-significant for this variable. Similarly, in mammalian viruses, host domestication status was also found to be non-significant due to a small sample size, which limited the predictive power of the

model[14]. In case of host body mass, a positive effect is expected as larger mammals can house a larger species richness of parasites[23,35]. However, very large and massive mammals may have smaller population sizes and isolated geographic ranges, therefore limiting parasite transmission. Our study agrees with this result, albeit this relationship was only marginally significant (Id=10.2 in Table 2), and this negative effect of host body mass on pathogen sharing has also been noticed in the literature[18]. Overall, our preferred model suggests that the relationships between the probability of host range expansions and various parasite and host properties, as well as abiotic factors, are stronger than previously thought.

With a sensitivity of 0.680 and specificity of 0.780 (Model Id=3 in Table 1), our model can effectively forecast the risk of a host shift in single-host parasites for mite species associated with hosts that frequently come into close contact with humans, such as captive and domesticated mammals. Risk assessment is also needed for invasive mammal species expanding their geographic ranges following climate change, habitat loss, or through direct human activities. These hosts can carry unwanted and potentially dangerous acarine parasites, which may shift to wild or domesticated animals and humans themselves[8].

In host-risk forecasting, our model identified rodents (Rodentia), bats (Chiroptera) and Carnivora as hosts harboring the highest number of risk-group mites i.e., potentially unobserved multi-host species (Fig. 3c; Table 3). As these three orders are species-rich, they provide more opportunities for host shifts among phylogenetically similar hosts (Fig. 4a). In addition, cross-species contacts promoting horizontal mite transmission are expected in these three host orders. In Carnivora, horizontal pathogen exchange can be facilitated through predation on various mammalian species which harbor an array of diverse acarine parasites forming natural associations with their hosts. In bats, mite pathogen exchange may be facilitated through the formation of large multi-species aggregations in the same roosting sites[36]. Put differently, these two host orders exhibit certain ecological characteristics that promote mite sharing.

When used for forecasting of risk-group mites, our preferred model identified 86 species at a probability threshold of 0.7 (Supplementary Data 1). Among these potentially multi-hosts parasites are five species of sarcoptid skin mites, *Notoedres ovatus* and *N. yunkeri*, *N. anisothrix*, *N. helicothrix*, and *N. eptesicus*. Currently, each of these species have been found on a different host species of molossid/vespertilionid bats, however our model predicts with a high probability, 0.866 − 0.930, that they are actually multi-hosts (Fig. 3b; Supplementary Data 1). There is some indirect evidence suggesting that host range expansion is plausible for these five single-host parasites. The genus *Notoedres* originated as a bat parasite, however, its crown group contains species resulting from historical host shifts from bats to other mammalian host orders, such as Rodentia (*Notoedres muris*, *N. musculi*) and Carnivora (*N. cati*)[37]. In other words, in the past, inter-ordinal host shifts followed by a speciation event occurred on a macroevolutionary scale in this genus. By analogy, one may conclude that these single-host mites predicted as multi-host by our model may potentially shift to other hosts and potentially cause an epidemic in new hosts. This scenario is likely assuming increased contact of molossid/vespertilionid bats and other mammalian lineages via predation and/or geographic range expansion via climate change or habitat disturbance.

In conclusion, we assembled the largest and the most complete dataset to date on mites permanently parasitic on mammals and developed a predictive model to analyze a set of determinants influencing the likelihood of single-host parasites transitioning into multi-host parasites. Our model accounted for potentially unobserved host-parasite links and class imbalances, identifying statistically significant predictors related to parasites, hosts, climate, and habitat disturbance. This analysis provided valuable insights into the ecological and epidemiological aspects of mammalian acarine parasites and potential disease transmission dynamics. When applied to forecast epidemic

risk-group parasites, our model revealed that rodents (Rodentia), bats (Chiroptera) and Carnivora harbor a disproportionately large number of single-host parasites with the potential to become multi-host, including the sarcoptid skin mites of the genus *Notoedres*, posing significant epidemic risks. Our study is one of the major attempts to analyze host specificity patterns in mammalian acarine parasites in a predictive, quantitative framework. However, more empirical and experimental studies are clearly needed to understand the general properties underpinning host-parasite interactions.

## Methods
### Mites
We selected acariform mites forming permanent (full-time) host associations with mammals (i.e., they do not have a free-living or specialized dispersal stage like chiggers or ticks) to be included in the database. Acariform mites include two major and distantly related lineages, Prostigmata and Astigmata, showing multiple independent origins of parasitism[38] and thus allowing phylogenetically independent comparisons, e.g., most prostigmatan families (n = 6) have colonized mammals independently, and astigmatan mammal mites are bi- or triphyletic[39]. Among these mites are relatively benign human-specific (single-host) symbionts, such as follicular mites, *Demodex folliculorum* and *D. brevis*[40]. *Demodex* mites are common components of healthy skin, acting as mostly harmless symbionts[41]. They have had a long history of co-evolution with mammals, and likely have mechanisms to evade and manipulate the host immune system[42]. In contrast to mostly non-pathogenic and single-host *Demodex*, the multi-host mite *Sarcoptes scabiei*, also belonging to an ancient lineage showing codivergence since the time of the marsupial/placental dichotomy[37], causes a highly contagious skin disease. This mite disease affects more than 200 million people, particularly in resource-poor tropical regions[43]. In immunocompromised humans or when host-shifted from humans to other mammals, the disease may develop into a crusted form, which is often lethal[44]. Multi-host skin mite parasites (*Otodectes*, *Psoroptes*, *Chorioptes*, *Sarcoptes*) pose threats to domesticated and wild mammals, including endangered species[44,45]. Because all developmental stages in our target taxa live on (or inside) the host body, there is a limited possibility for dispersal or host shifts other than by direct body contact, such as mother to offspring vertical transmission, mating, or within-species social contact[46]. However, interspecific (among-species) mite transmission may be facilitated by hosts sharing the same habitat (e.g., bat roosting sites) or predator-prey interactions (e.g., hyaenas and hedgehogs) (our data). Because these mites lack a vector transmission, our analysis should not be confounded by an additional layer of complexity associated with traits of vector organisms.

### Host-parasite database
Using the literature, particularly recent reviews[47,48], we compiled a taxonomic database of parasitic acariform mites, including the following information: (1) valid mite name, authority, and year; (2) unique host records per mite species; (3) mite taxonomy (family, parvorder); (4) biogeographic region; and (5) credibility of host record (see below). Our database has 3489 unique host-parasite records representing 1998 mite and 1486 mammalian species (Supplementary Data 3). Based on interviewing authors about their sampling procedures and the repeatability of a particular association, we marked 113 records as potential sample cross-contamination in the field, laboratory or museum (low credibility). After quality control (e.g., missing data, uncertainty in host identification, low credibility records) and synchronizing the host taxonomy with the latest source, the Mammal Diversity Database (MDD) v1.10[49], our dataset had 3350 unique host-parasite records representing 1,984 mite species (22 families, 2 parvorders) and 1432 mammal species (118 families, 22 orders). A host name lookup table was created to retrieve data from four external sources, all using different host taxonomies: host classification,

domestication status, and biogeographic region (MDD v1.10)[49]; host phylogeny[50]; host traits and environmental data (PanTHERIA)[51]; shape files describing spatial distribution of mammalian hosts (MDD v1.2)[52]; and Google Scholar publication count per mite species accessed on May 20 2023 (custom Python script: see Code Availability). After removing observations with missing data (mostly due to external host databases) and after summarizing our final dataset by unique mite species (as required by downstream analyses), our analysis dataset had 1445 unique observations (Supplementary Data 4).

### Dependent variable
The variable *multihost* has two categories describing the mite host range, either No (*singlehost*, 0) or Yes (*multihost*, 1). There were 1032 (71. 4%) single-host and 413 (28.6%) multi-host mite species, indicating that our dataset is moderately imbalanced with respect to single-host mite species, which is the majority category. Therefore, when the imbalanced nature of host-parasite association is not accounted for, a random mite record has a higher probability of being classified as single-host.

### Predictors
We coded 14 variables belonging to three general groups: (i) mite-related: *Host Immunity Contact Level*, *Chelicerae*, *Parasitism*, *Mite Bioregion*, *Mite Dispersal Stage*, *Precopulatory Guarding*; (ii) host-related: *Co-Distributed Potential Hosts* (see below), *Potential Phylogenetically Similar Hosts* (see below), *Average Host Litter Size*, *Domesticated Host*, *Average Host Body Mass log*; (iii) climatic: *Average Precipitation*, *Average Temperature*, and (iv) habitat disturbance: *Average Human Population Density log*. The mite-related predictors were coded using our data and the following main sources[38,53–55]. *Co-Distributed Potential Hosts* and *Phylogenetically Similar Potential Hosts* were calculated using a custom script (see below), while the remaining host-related and environmental variables were imported from the PanTHERIA database[51]. Our analysis is a mite species-level analysis, i.e., each mite species (or subspecies) has a set of values related to the mite itself, host(s), or the environment. In the case of multi-host mites, we averaged the values of corresponding host-related predictor variables to ensure that the combined effect of all hosts is appropriately represented. For two variables, *Co-Distributed Potential Hosts* and *Phylogenetically Similar Potential Hosts*, we used maximum values because the effect of these variables is expected to be most pronounced at maximum values, which will be explained in the following section.

Two variables, *Host Immunity Contact Level* and *Chelicerae*, were nearly perfectly correlated (see below), so only the former was used in downstream analyses. Therefore, our final dataset consisted of the following 13 predictor variables, with variable names italicized for readability:

1. *Host Immunity Contact Level* describes the three levels of contact with the host immunity: (1) *Primary*—mites directly contact elements of the host immune system while feeding; (2) *Secondary*—mites feed superficially on outer (dead) epidermal tissue; immune response occurs subsequently, usually because of a combination of host abrasion/scratching and mite presence, resulting in oozing inflammatory lipid exudates and lymph secretions (which can also be consumed by mites); (3) *None*—mites feed on host tissue derivatives with no immune response, e.g., sebaceous secretions. For example, myobiid mites pierce the skin and feed on the lymph and intercellular fluids of the host[56] (*Primary*). The skin mite *Psoroptes ovis* abrades the stratum corneum, depositing allergens as they progress, and this combination of skin abrasion, allergen deposition and self-grooming behavior by the host in response to the pruritis triggers the subsequent activation of a cutaneous inflammatory response[53,54] (*Secondary*). Fur mites (Listrophoridae, Atopomelidae, Chirodiscidae) feed on materials accumulated on the hair surface: shed dead skin, sebaceous gland secretions, fungal spores, hyphae, and pollen[57] (*None*). Mites with the

states *Secondary* or *None* are expected to have a higher likelihood of establishing on a new host species (host range is broader) as they do not have direct contact with the host immune system, while mites with the state *Primary* are expected to have narrower host ranges. The variable *Chelicerae* describes the shape of the mite chelicerae (a mouthpart): *chelate* or *piercing*. All prostigmatan mites treated here have piercing chelicerae, while psoroptidid astigmatan mites, with the exception of the family Lemurnyssidae, have chelate chelicerae. The values *Host Immunity Contact Level*: *Primary* and *Piercing Chelicerae* are perfectly correlated with each other; if *Host Immunity Contact Level* is *Secondary* or *None* then chelicerae are always chelate. Thus, to avoid collinearity, the variable *Chelicerae* was excluded from further analyses. It should be noted that due to the nature of statistical data, predictor variables are not expected to perfectly separate the groups of the dependent variable. For example, even though mites with chelate chelicerae are expected to be multi-host, a small proportion of mites with chelate chelicerae may still be strictly host-specific (single-host), such as several species of the genus *Chirobia*[37].

2. *Parasitism* describes whether the mite is ectoparasitic (lives outside the host body) or endoparasitic (lives inside the host body). Examples are fur mites living on the host hair (*Ectoparasitic*) and species of Gastronyssidae and Ereynetidae living in the nasal cavities of their hosts (*Endoparasitic*). Ectoparasitic mites are expected to have a higher likelihood of being transferred on contact between different host species (broader host range) than endoparasitic mites (narrower host range). The four species-level taxa of *Opsonyssus* (*O. brutsaerti indica*, *O. pseudoindicus*, *O. eidoloni*, *O. pteropodi*) could be considered both endoparasites and ectoparasites due to their habitat on the outer surface of host eyeballs. Here, we conservatively coded them as ectoparasitic. Our sensitivity analysis using alternative coding (*Endoparasitic*) yielded very similar results in terms of coefficient estimates and classification accuracy (Supplementary Table 2), suggesting that our model is robust to different interpretations of the parasitism type in *Opsonyssus*.

3. *BiogeographicRegionMite* describes the known geographic distributions of mite species (not to be confused with host geographic distribution). We used several biogeographic coding schemes (MA=Madagascar, PM=Papua New Guinea, SA=Sahara Desert): 10 categories (Afrotropic, Afrotropic_MA, Australasia, Australasia_PM, Ethiopian_SA, Indomalaya, Nearctic, Neotropic, Palearctic, *Multiregion*), 6 categories (Afrotropic, Australasia, Holarctic, Indomalaya, Neotropic, *Multiregion*), and 2 categories (*Singleregion*, *Multiregion*). Multiregion refers to mites that are distributed in more than one biogeographic zone. We used the 2-category scheme in our final analyses because the 10- or 6-category schemes did not significantly improve the group separation (single- vs. multi-host mites) in the dependent variable. Mites having the *Multiregion* state are expected to have wider host ranges because they have adapted to a wider range of environments and have higher probabilities of being exposed to higher diversity of host species, while the reverse is expected for *Singleregion* mites.

4. *Mite Dispersal Stage* refers to the specific life stage during which mites primarily disperse (in contrast, adult males are always mobile and are not categorized within this variable). In most mite lineages, dispersal occurs through all life stages, including adult females and immatures (*F_Imm*), which are capable of walking on the host. However, in several mite groups, females may have modified characteristics rendering them sedentary, while dispersal is primarily achieved by immature stages, often just larvae (*Imm*). For example, in the genus *Gastronyssus*, females live in the host's stomach and are elongated, while larvae are found in the mouth and nasal cavities, indicating that this immature stage is the dispersal stage[58]. Similarly, in the sarcoptid genera *Rousettocoptes*, *Tychosarcoptes*, *Chirobia*, *Teinocoptes*, females are physogastric (extreme enlargement of the body) and cannot move, while dispersal is accomplished by larvae[37]. Finally, some lineages

disperse only as adult females (*F*). For example, immature female stages of the chirodiscid subfamily Labidocarpinae lack functional legs and only acquire them at the final molt, when they become adult females[59]. Even though the legless labidocarpine immatures cannot disperse on their own, they still can be dispersed by males as part of precopulatory guarding behavior (see below). Mites that exhibit dispersal across all stages (*F_Imm*) may achieve greater establishment success, suggesting that species possessing this trait are likely to have broader host ranges, while species having a restricted set of dispersal stages (*Imm*) or (*F*) are expected to have narrower host ranges.

5. *Precopulatory Guarding* describes the presence or absence (*y/n*) of precopulatory guarding, i.e., an adult male grasps an immature female (unreceptive or nymphal) and carries it for an extended period of time prior to mating[60]. The presence of precopulatory guarding increases the likelihood of a successful transmission since both sexes can be transmitted to a host individual in a single colonization event (wider host range), while species exhibiting no precopulatory guarding are expected to have narrower host ranges.

6. *Co-Distributed Potential Hosts* (*n_CPH_max*) characterizes the number of mammal species that have spatial overlap with the host/hosts associated with a mite species. To calculate this quantity, we first computed an m-by-m matrix of all pairwise geographic overlap of mammals that have distribution information in the MDD v.1.10 database[49], where 'm' indicates the total number of mammalian host species. Geographic overlap was assessed using pairwise Jaccard Similarity Coefficient,

$$J = \frac{G_A \cap G_B}{G_A \cup G_B} = \frac{Area_{overlap}}{Area_A + Area_B - Area_{overlap}} \tag{1}$$

To calculate the *n_CPH_max* for each mite species, we followed these steps: (1) for a single-host mite, we matched the corresponding matrix row representing its host and determined *n_CPH_max* by summing the number of Jaccard similarity values exceeding 0.5; (2) for a multi-host mite, we repeated step 1 for each host and retained the maximum count. The *n_CPH_max* measure provides more probabilistic metrics to estimate the available potential hosts in the mites' geographic distribution. We used the maximum count because that host serves as the most likely reservoir of parasites for new hosts (see Code Availability for custom code).

7. *Potential Phylogenetically Similar Hosts* (*n_PSH10_max*) characterizes the number of mammal species that are phylogenetically closely related to the parasite host/hosts. It was calculated in a similar manner to *n_CPH_max*, but using pairwise phylogenetic distances between mammal pairs from the latest mammal supertree[50]. The *n_PSH10_max* measure counts the number of mammal species whose phylogenetic distances to the parasite host are smaller than 10. For multi-host mites, *n_PSH10_max* is chosen to be the maximum count among the hosts. This measure provides a probabilistic metric to estimate all the available potential hosts on the mammal tree that are phylogenetically similar. Unlike raw phylogenetic distances and related metrics, our metric is agnostic to the current parasite's host range status, particularly whether if the parasite is single- or multi-host (see Code Availability for custom code).

8. *Average Host Litter Size* describes the number of host offspring born per litter per female; PanTHERIA: *15-1_LitterSize*[51]. We expect that larger litter sizes increase the parasite dispersal through the vertical route of transmission (parent to offspring) and, in addition, large litter sizes may be correlated with host fitness and the propensity to support large parasite loads[30]. Hence, this variable is likely to be positively correlated with the elevated probability of host range expansion.

9. *Domesticated Host* (*y*, *n*) accounts for empirical observations that pathogens associated with domesticated animals experience frequent host switches resulting from the increased opportunity for cross-infestation in anthropogenic settings[10,61,62]. Here, we used a list of

domesticated hosts as defined in MDD v.1.10[49], amended with common peri-domestic pest animals, such as the black rat *Rattus rattus* and humans themselves, which are often the source of mites secondarily infesting domestic animals[63].

10. *Average Host Body Mass log* describes the host body mass (g); extracted from the PanTHERIA database, variable *5-1 AdultBodyMass_g*[51]. In our analyses, raw values were converted to natural logarithms. We expect nonlinear/bi-directional relationships with the dependent variable: in the low to middle value range of this variable, larger mammals may harbor larger population sizes of parasites, facilitating transmission, and therefore promoting the evolution of multi-host parasites (positive correlation with the dependent). However, very large and massive mammals may themselves have smaller population sizes and isolated geographic ranges, therefore limiting parasite transmission (negative correlation with the dependent). When applied to massive animals, this negative effect of host body mass was noted in the literature[51].

11. *Average Precipitation* is the mean monthly precipitation (mm) within the geographic range of the host; PanTHERIA: *28-1_Precip_Mean_mm*[51]. We expect that precipitation positively affects the mite transmission probability as mites are prone to desiccation while outside the host − higher humidity extends the short period when mites can survive in the environment.

12. *Average Temperature*, mean monthly temperature (x0.1 °C) within the geographic range of the host; PanTHERIA: *28−2_Temp_Mean_01degC*[51]. This variable is expected to be negatively correlated with the dependent as higher temperatures may promote mite desiccation and decrease survival while being transmitted across hosts.

13. *Average Human Population Density log*, 5th percentile human population density (persons per km2); PanTHERIA: *27-3_HuPopDen_5p_n/km2*[51]. This variable is a proxy for anthropogenic habitat disturbance. Environmental disturbance is expected to promote host switches (host range expansion) as there are increased opportunities to encounter new hosts[64].

## Host range model

To determine which factors affect host range expansion in a single-host mite, we ran a logistic regression using Generalized Linear Model (*glm*) in *caret*[65], with some of the continuous variables transformed into polynomial variables using natural spline function in *splines* for R v4.2.2[66]. Natural splines can fit smooth regression curves to nonlinear data, with the number of spline curves controlled by the user through the degree of freedom parameter or set automatically via cross-validation. Generalized Additive Model (*gam*) is similar to Spline Regression, but it selects parameters for smooth basis functions automatically, sometimes leading to overfitting / unrealistically wiggly smooth functions, especially in areas with little data[67]. We could not alleviate this issue by increasing the basis dimension parameter (k) and, therefore, use Natural Spline Regression over *gam* here. After the removal of 19 records from captive hosts (natural host distribution is unknown), we randomly split our dataset into train (*n* = 1024, 70.9%) and test (*n* = 421, 29.1%) subsets. Because our data had disparate frequencies of the observed host range classes (class imbalance: single-host 71.4%, multi-host 28.6%), we simultaneously subsampled (up- and down-sampling) and resampled (5-fold cross-validations, 5 repeats) our train dataset to estimate the performance of different models based on AUC-ROC (area under the Receiver Operating Characteristic curve) metric in *caret*[65]. For comparison, we used the baseline unweighted model, a weighted model, and a positive-unlabeled (PU) learning model with AdaSampling[68].

The weighted model weights records by Google Scholar publication counts for each mite species as follows: 0.2 + (totalPubs > =10) *0.8. This model accounts for the fact that some mite species received more research efforts than others, and thus their records of association with hosts are more complete than other species. This formula gives full weights to mite species with more than 10 publications on Google Scholar and 0.2 weights to the species that have less than 10 publications. This weighting scheme was designed to mitigate the impact of unobserved multi-hosts, which are likely to be associated with a lower sampling effort, hence having lower publication counts.

A positive-unlabeled (PU) learning model explicitly assumes that labels in the positive class are known while the negative class is unlabeled and contains a mixture of both positive and negative instances[69]. This contrasts with traditional supervised learning (such as spline regression), where both positive and negative instances are assumed to be accurately labeled. In our case, the positive class is multi-host, which should be correctly labeled because once a mite species is credibly classified as multi-host, the multi-host status of this mite species will remain unchanged with additional sampling. In contrast, the negative class is unlabeled and contains both true single-hosts and true multi-hosts erroneously classified as single-hosts due to insufficient sampling (unobserved multi-hosts). PU learning algorithms typically aim to estimate the probability that an unlabeled instance belongs to the positive class, leveraging information from the labeled positive instances and the unlabeled data[69].

For model selection, we used several metrics: ROC-AUC, Accuracy, Kappa, Sensitivity, Specificity, and F1. Of them, the former three provide a measure of overall model performance (still can be biased to class imbalance, especially Accuracy), while the latter three evaluate the positive class performance (*multihost*). For our main metric, ROC-AUC, we evaluated 95% confidence intervals in the R package *pROC* integrated with *caret*. We used our final model to predict the test (holdout) dataset, i.e., records that were not used in model inference, and further analyzed records observed as *singlehost* but predicted as *multihost* by the model. Mites in this category have the highest risk probability to become multi-host (high host switching / host expansion risk) given the data. We then calculated the 'enrichment' of the multi-host risk group among different mammal orders for each host order by subtracting the percentages of observed *singlehost* which were predicted as *multihost* by the model at different classification thresholds; from these values, we subtracted the percentages of observed *singlehost* records per host order. Values above zero (above the average) are considered as high-risk single-host parasites that are likely to become multi-hosts.

## Mite-sharing model

The n-by-n mite-sharing matrix of hosts was constructed by projecting mite-host association records into a unipartite network, where an edge between a host pair represents the sharing of one or more parasites. Hosts that do not share any parasites with other hosts were excluded from the analyses. The final dataset was constructed by taking the intersection of host species that were present in all three matrices: the n-by-n mite-sharing matrix and the m-by-m matrices of all pairwise host geographic overlaps and phylogenetic distances (see above), and sub-setting the three matrices accordingly to be of the same size. The distribution of pairwise phylogenetic distances (PD) is very uneven due to the deep divergence between marsupial and placental mammals (PD > 300). We recoded these PDs to be 178 (i.e., the highest PD within placental mammals) so that models would not overfit the gap region between large PDs. We discretized the PD and geographic overlap (GO) and counted the cases of observations in each combination. 91% of the host pairs (833,454 out of 915,981 pairs) do not have geographic overlaps and are phylogenetically very divergent from each other (Fig. 4a, b, Supplementary Fig. 2 a, b). To avoid overfitting in the regions of extreme values, we sub-sampled each bin down to 1000 pairs (Supplementary Fig. 2 b). We used a generalized linear model in which the probability of observing mites sharing between hosts is a

linear response of GO, natural splines of PD and their interactions. Specifically,

$$\text{Logit(MitesShare)} \sim ns(PD,2) + GO,1 + PD{:}GO \qquad (2)$$

Using this model, we can predict the probability of mites sharing between any pair of mammal hosts given their PD and GO. The statistics of the model output was summarized in Supplementary Table 1. Our entire statistical pipeline is documented through R scripts (see code availability).

### Reporting summary

Further information on research design is available in the Nature Portfolio Reporting Summary linked to this article.

## Data availability

All data generated in this study, including the main host-parasite database, are provided in Supplementary Data and have been deposited on the Zenodo database at https://doi.org/10.5281/zenodo.11130648[70]. See Methods for data sources used to compile the host-parasite database. All the source data for generating the figures are provided in Supplementary Information/Source Data File. Source data are provided with this paper.

## Code availability

The scripts for data analyses and prediction are available on Zenodo at https://doi.org/10.5281/zenodo.11130648[70].

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

## Acknowledgements

We thank Barry OConnor (University of Michigan, Ann Arbor, USA), Sergey Mironov, and the late Andrey Bochkov (Zoological Institute, Russian Academy of Sciences, Saint Petersburg, Russia) for comments and discussion. We also thank four anonymous reviewers for their insightful suggestions, which helped us improve the earlier versions of the manuscript.

## Author contributions

P.B.K. and Q.H. conceived the overall research plan, designed the experiments, analyzed the data, and wrote the manuscript. P.B.K. generated the host-parasite dataset.

## Competing interests

The authors declare no competing interests.
