## [Peer Review File · Nature Communications]

REVIEWER COMMENTS

Reviewer #1 (Remarks to the Author):

This is a welcome, and audacious, attempt address a major issue, the probability of specific parasites to switch hosts. As noted by the authors, greater ability to switch host species is often associated with greater virulence and thus predicting host switching probability is of considerable practical importance. The authors use a large data set of parasitic mite species and a very extensive set of variables, generating the most comprehensive analysis of this issue I am aware of. In the process they introduce some very much needed rigor into this area, allowing testing of some of the many hypotheses proposed. All of these very positive.

In terms of the analysis techniques, I lack the expertise to comments on the details of the statistical analyses, but the explicit incorporation of techniques to account for class imbalance is very welcome. The conclusion that the factors "host that is spatially co-occurring with other potential hosts ..." and "contact with the host immune system" are the most important factors determining host switching success is very gratifying. This has been assumed/hypothesized before, but this study provides some statistical back-up. Adaptation to hosts (the consideration involved in "contact with the host immune system") is most commonly invoked in discussions of evolution of host associations. It is undoubtedly important, although of different levels of importance in different lineages, but it may not always be the most important factor. Opportunity to transfer, as determined by dispersal mode (mostly direct body contact in these groups) and overlap of host (micro-)habitats ("host that is spatially co-occurring with other potential hosts ...") is absolutely critical. I would have guessed that it would not be easy to model, but the analyses did identify it as one of the two most important factors. Quite impressive.

My criticisms are largely minor items that are always associated with large analyses. Much of this is informed by my knowledge of especially the family Sarcoptidae.

The scoring procedures

1. One of the major complications in this type of analysis is that scoring has to be with discrete, either/or, characters. This does not always reflect the reality of intermediate conditions for some scored factors. For example, are *Opsonyssus* (Gastronyssidae) species moving around on the outer surface of the eyeball endo-or ectoparasites? I cannot see a way of fixing this, but it might be worthwhile to acknowledge it.
2. Not all potentially important variables have been scored. For example, in Sarcoptidae, probability for successful host switching is almost certainly affected by the nature of the dispersing instar(s): females in Sarcoptinae, immatures in Notoedrinae. This is probably most strongly expressed in teinocoptines (Notoedrinae) where females and nymphs are highly modified, making it extremely difficult for them to walk at all, leaving dispersal almost exclusively to larvae (and males)). Dispersal by larvae / immatures would be expected to be less efficient than dispersal by (pregnant) females.
3. Both geographical overlap and phylogenetic distance are likely to affect transfer success. It is not quite clear why the two are combined.
4. Authors argue that the character piercing vs. chelate mouthparts overlaps with, respectively, single host and multi-host. Those statements need to be weakened a little. Even ignoring non-mammal associated parasites (e.g. *Tetranychus urticae* with piercing mouthparts on many plant species), the statement is incorrect for some members of the test group, e.g. in Sarcoptidae. Members of the genus

Chirobia have chelate mouthparts and appear quite host specific.

5. Grammatical and minor issues

- a) L 306-308 " Demodex mites are common components of healthy skin, have a long history of co-evolution with mammals, and likely have mechanisms to evade and manipulate the host immune system." A correct statement, but Sarcoptidae also have a history of association with mammals since the time of the marsupial/placental divergence. Yet Sarcoptidae maintain the ability to switch to new mammalian hosts, even to new orders. Based on that, the statement " have a long history of co-evolution with mammals" may not be relevant.
- b) L 158 please add unit to 140 (mm/year?)
- c) L 158 eliminate ", which"
- d) L 158 eliminate "and"
- e) L174 it IS STILL POSSIBLE
- f) L 245 eliminate "a" after lacking
- g) L 251 "while only a few predictors were related to hosts". How about factor PredictedMitesShare, the most impactful factor. It is host related (L 356). It should be mentioned here.
- h) L 276 eliminate "two"
- i) L 276 how about "the genus Notoedres originated as bat parasites"

Miscellaneous comments. Successful host switches, leading to establishment on a new host species) are probably not very common in natural situations. Even switches to new host individuals of the same species may require immune-compromised hosts (infections of healthy hosts may be self-limiting in e.g. Sarcoptes). As also noted by the authors, most switches end in the death of the parasite. I do wonder if there would be worthwhile to try to model overall switching probability vs. successful switches (no idea how to do that).

All of the above is largely nit-picking and most likely will not influence the results of the analysis. Overall this is a very worthwhile study.

Reviewer #2 (Remarks to the Author):

I would like to congratulate the authors on conducting a comprehensive and insightful study on a topic of major importance. The assembly of such a large and complete dataset to analyze the factors affecting the transition of single-host parasites to multi-hosts is indeed a commendable effort. The study provides a robust quantitative framework that can potentially guide future research in this area. I have no major comments; here are just a few comments and suggestions that might help in further refining the manuscript:

Abstract:

Line 17-19: To provide a clearer context right from the beginning, I would suggest a minor edit in the phrase "When single-host mites are". It could be revised to "When parasitic single-host mites are" to emphasize the parasitic nature of the mites being discussed.

Line 27-30: In the section where the forecasting of Chiroptera (bats) and Carnivora as hosts with a high

number of parasites in the multi-host risk group category is mentioned, it might be beneficial to elaborate slightly on why this result is significant and how it integrates with the broader findings of the study.

Line 29-30: The phrase "that are probably capable of causing an epidemic". It might be more informative to specify the kind of epidemic that is being referred to here, to give readers a clearer picture of the potential implications.

Main Text:

Line 39-61: Given that host specificity is central to this study, incorporating a discussion or mentioning the widely recognized three facets of host specificity (raw, phylogenetic, and geographic), as considered in the study by Poulin, R., Krasnov, B. R., & Mouillot, D. (2011), might be beneficial.

Material and Methods:

Line 495: It would be great to see a bit more detail regarding the rationale behind the formula used in this section.

Figures:

Figure 2b: To enhance the visual representation, it might be beneficial to order the data from higher to lower values, facilitating a more intuitive understanding of the data distribution.

Reviewer #3 (Remarks to the Author):

This manuscript models the probability of single-host (or specialist) mites becoming multi-host (or generalist) parasites. The study uses an impressive dataset of mite-host associations and accounts for many variables that could potentially play a role in the shift from single-host to multi-host. The details of the modeling approach are outside my expertise, but overall I think the study is thorough and the conclusions are sound based on the methods and the results. I found the figures to be informative and easy to interpret. I think this research is a valuable contribution to understanding host-parasite interactions and can inform the associations formed by other parasites. I have no major critiques, but do have some questions and feedback noted below.

Main Section

I think it would be helpful to include a little more information about the mite, host, and abiotic traits that can influence if a mite is single-host or multi-host. It doesn't need to be a thorough explanation, but a bit of background would be helpful to understand the variables as they are explained in the "Factors affecting the odds to become multi-host" section.

Discussion Section

Lines 271–276: there is a list of four species of sarcoptid skin mites, the following sentence refers to "these two single-host parasites" and it isn't clear what two that statement is referring to.

Lines 279–280: "In other words, in the past, inter-ordinal . . ." something about the wording of that sentence is not correct. I think it may just be missing "by" between "followed" and "a", but I'm not

certain.

Methods Section

Line 350: I think that it should be "host-parasite association" instead of "host parasite-association"

Lines 394–401: This section explains the biogeographic regions and the different coding schemes, but it does not explain why the last scheme (singleregion and multiregion) were used in the analyses. Why was that scheme used?

Reviewer #4 (Remarks to the Author):

Review for Klimov and He

This paper addresses an interesting question in assessing the predictors of multi-host capacity in mites, with the added contribution of assembling a large dataset of host-mite associations. It's a valiant attempt at these complex analyses with some good aspects to the paper, but altogether it comes across as relatively hastily conducted and with many gaps in construction, interpretation, and presentation. As such, I can't recommend its publication. Apologies for not being more positive. Some of my more major comments are below.

On the positive side:

- The database is a very useful contribution to the literature on host-parasite associations.
- The models are interesting and advanced.
- The figures are aesthetically pleasing and apparently extremely well made and indicative of strong analytical expertise.

Misc:

- The title is far too general, and should absolutely not be allowed to be published as it is. This is a paper about mites, and the title should be restricted to reflect that. The title gives the impression that it is a paper about parasites in general, and a review of drivers rather than an analysis, and gives no indication of what was actually found.
- The structure of the discussion and results are difficult to tie together when reading, and the changeable and imprecise language used makes it difficult to link them to each result in the abstract.

Approach:

- The observation that a given mite has only been seen on members of one species does not necessarily make it a "single-host" parasite, as there may just be unobserved hosts, as the authors themselves acknowledge (line 210-213). At best, the analysis is picking up a reduced mean host range that is reflected more often at the species level as an observation of only one known host in a number of species. The number (and proportion) of actual "single-host" specialists is completely unknown, as far as I am aware. I am very doubtful that a subsampling approach like the one described could manage to overcome this data limitation. Feeding data into a reclassification model based on that same dataset and then restricting it and analysing a subset provides (in my opinion) false hope that these data hurdles have been overcome.

- If the analysis was phrased as “factors contributing to host range in mites” rather than “switching from one to multi host” that would solve a lot of the approach-related problems the paper suffers from, and the models could stay much the same as long as the interpretation was modified in the writing and appropriate attention was given to sampling bias. I’m confused why the authors approached it in this binary way rather than simply proceeding with the host count data they have.
- The predictors are relatively underpresented in the introduction in terms of hypotheses for why specific mite and host traits should create variation in host range.

Model:

- Combining two variables (phylogenetic distance and geographic overlap) into one explanatory variable is an odd and unjustified approach that could do extremely strange things that would be avoided entirely by including both variables independently. I think this was carried out because the analysis as designed required a single variable? But why could these variables not be included in the analyses as they were?
- The analysis is a mite-level analysis I think (the unit of replication is the mite, where each species is a unique value. But a lot of these variables are related to the host, and it’s unclear how each value was reached. Some variables it’s explicitly max, some it’s mean, and others give no info. Or was the unit of replication a mite-host association, such that each host was represented multiple times, and I’ve misunderstood? More clarity needed.
- The sharing model itself is very interesting and I’m not sure why it was so sidelined and reduced to producing an explanatory variable in another model.

Writing:

- The paper is fairly inaccurately written throughout: for example, on line 25, what is “the proximity to the host immune system”?
 - The discussion is roundabout and poorly tied to the authors’ results.
 - Related, the grammar needs checking thoroughly. E.g. line 75 “acariform mites mite-mammal-associations” is fairly sloppily written. Line 77, “all known target parasites and their all known hosts”, Etc.
 - Beyond this, it is also excessively jargony in points – the paragraph beginning line 133 has a lot of GAMM-specific wording that would be very difficult to dissect for an unfamiliar reader particularly if they haven’t yet read the methods.
 - The analysis is very complex, and the “results first” format is doing it no favours. Lots of variables and results come out of the blue with no context, making it very difficult to follow the thread of importance.
 - The reader is left with relatively little idea of what to take from the paper on finishing the discussion; this analysis is doing a lot of different things, and none of them is given sufficient time or attention, such that nothing in particular shines through.
- 68-70: I’m not sure this assertion is either true or addressed by this paper.
 - 150-153: Why interpret a non-significant result? This interpretation should be in the discussion, anyway, not the results. Same with the marginal result at 170-172.
 - 248-250: I’m no closer to understanding this variable (which seems to combine two very different traits) than I was in the abstract. Having read the methods I now get it, but it needs far more context adding where it’s mentioned before the methods.

Response to Referees

Our point-by-point responses to reviewers' comments are provided below. The primary modification in our modeling approach is the incorporation of a positive-unlabeled analysis, which fully addresses the concerns raised by reviewer 4. Please refer to R4.A3 for details.

REVIEWER COMMENTS

Reviewer #1 (Remarks to the Author):

This is a welcome, and audacious, attempt address a major issue, the probability of specific parasites to switch hosts. As noted by the authors, greater ability to switch host species is often associated with greater virulence and thus predicting host switching probability is of considerable practical importance. The authors use a large data set of parasitic mite species and a very extensive set of variables, generating the most comprehensive analysis of this issue I am aware of. In the process they introduce some very much needed rigor into this area, allowing testing of some of the many hypotheses proposed. All of these very positive. In terms of the analysis techniques, I lack the expertise to comments on the details of the statistical analyses, but the explicit incorporation of techniques to account for class imbalance is very welcome.

The conclusion that the factors "host that is spatially co-occurring with other potential hosts ..." and "contact with the host immune system" are the most important factors determining host switching success is very gratifying. This has been assumed/hypothesized before, but this study provides some statistical back-up. Adaptation to hosts (the consideration involved in "contact with the host immune system") is most commonly invoked in discussions of evolution of host associations. It is undoubtedly important, although of different levels of importance in different lineages, but it may not always be the most important factor. Opportunity to transfer, as determined by dispersal mode (mostly direct body contact in these groups) and overlap of host (micro-)habitats ("host that is spatially co-occurring with other potential hosts ...") is absolutely critical. I would have guessed that it would not be easy to model, but the analyses did identify it as one of the two most important factors. Quite impressive.

My criticisms are largely minor items that are always associated with large analyses. Much of this is informed by my knowledge of especially the family Sarcoptidae.

The scoring procedures

R1.Q1. One of the major complications in this type of analysis is that scoring has to be with discrete, either/or, characters. This does not always reflect the reality of intermediate conditions for some scored factors. For example, are *Opsonyssus* (Gastronyssidae) species moving around on the outer surface of the eyeball endo-or ectoparasites? I cannot see a way of fixing this, but it might be worthwhile to acknowledge it.

R1.A1. In multivariate statistics, there is no straightforward method to incorporate ambiguous states, except for doing a sensitivity analysis to see if alternative coding changes results. We conducted a sensitivity analysis to explore whether alternative coding schemes would affect our results. In the manuscript, we have acknowledged this aspect as follows:

"The four species-level taxa of *Opsonyssus* (*O. brutsaerti indica*, *O. pseudoindicus*, *O. eidoloni*, *O. pteropodi*) could be considered both endoparasites and ectoparasites due to their habitat on the outer surface of host eyeballs. Here, we conservatively coded them as ectoparasitic. Our sensitivity analysis using alternative coding (*Endoparasitic*) yielded very similar results in terms of coefficient estimates and classification accuracy (Extended Data Table 2), suggesting that our model is robust to different interpretations of the parasitism type in *Opsonyssus*."

R1.Q2. Not all potentially important variables have been scored. For example, in Sarcoptidae, probability for successful host switching is almost certainly affected by the nature of the dispersing instar(s): females in Sarcoptinae, immatures in Notoedrinae. This is probably most strongly expressed in teinocoptines (Notoedrinae) where females and nymphs are highly modified, making it extremely difficult for them to walk at all, leaving dispersal almost exclusively to larvae (and males)). Dispersal by larvae / immatures would be expected to be less efficient than dispersal by (pregnant) females.

R1.A2. As suggested by the reviewer, we added a new variable *MiteDispersalStage* (with 3 levels: IMM, F, F_IMM) and updated the relevant sections of the manuscript (M&M, results, discussion, figures, supplements). See the manuscript text directly. Our model revealed the directional effect as suggested by the reviewer, i.e., dispersal by immatures is less effective, thus indicating a higher likelihood of being single-host. However, this effect was not significant in our model.

R1.Q3. Both geographical overlap and phylogenetic distance are likely to affect transfer success. It is not quite clear why the two are combined.

R1.A3 (see See also R4.Q6). We agree that it could be difficult to understand the individual influence of host geographical overlap and phylogenetic distance when coded as a combined variable. We therefore conducted a new analysis where variables related to host geographical overlap (*n_CPH_max*) and phylogenetic distances (*n_PSH10_max*) were included as separate variables. Our new analysis showed that both these variables are important and statistically significant predictors. We updated the relevant sections of the manuscript (Methods & Materials, Results, Discussion, Figures, Supplements) accordingly; please refer to the manuscript text directly for details.

R1.Q4. Authors argue that the character piercing vs. chelate mouthparts overlaps with, respectively, single host and multi-host. Those statements need to be weakened a little. Even ignoring non-mammal associated parasites (e.g. *Tetranychus urticae* with piercing mouthparts on many plant species), the statement is incorrect for some members of the test group, e.g. in Sarcoptidae. Members of the genus *Chirobia* have chelate mouthparts and appear quite host specific.

R1.A4. This is a misunderstanding. We did not argue that piercing vs. chelate mouthparts perfectly predict ("overlap with") single hosts and multi-hosts. Instead, we excluded this variable entirely from the analysis because it perfectly correlates with another variable, "Contact Level with Host Immune System".

With all due respect to the reviewer, we also note that the genus *Chirobia* appears to be not entirely host-specific. In this genus, five species are host species-specific (*Chirobia thoopterus*, *C. haplonycteris*, *C. jagori*, *C. minor*, *C. rousettus*), while six species have been recorded as multi-host parasites (*Chirobia otophaga*, *C. brevior*, *C. cynopteri*, *C. eonycteris*, *C. angolensis*, *C. squamata*) (Klompen, 1992).

To address the reviewer comment and eliminate the misunderstanding, we added the following sentence: "It should be noted that due to the nature of statistical data, predictor variables are

not expected to perfectly separate the groups of the dependent variable. For example, even though mites with chelate chelicerae are expected to be multi-host, a small proportion of mites with chelate chelicerae may still be strictly host-specific (single-host), such as several species of the genus *Chirobia*⁴⁰.

Klompen, J.S.H., 1992. Phylogenetic relationships in the mite family Sarcoptidae (Acari: Astigmata). Miscellaneous Publications Museum of Zoology University of Michigan 180, 1-154.

Grammatical and minor issues

R1.Q5a L 306-308 "Demodex mites are common components of healthy skin, have a long history of co-evolution with mammals, and likely have mechanisms to evade and manipulate the host immune system." A correct statement, but Sarcoptidae also have a history of association with mammals since the time of the marsupial/placental divergence. Yet Sarcoptidae maintain the ability to switch to new mammalian hosts, even to new orders. Based on that, the statement "have a long history of co-evolution with mammals" may not be relevant.

R1.A5a. We rewrote this text to make it clear that we contrast single-host mites (which usually are less pathogenic) and multi-host mites (which usually are pathogenic).

Old text: "The multi-host mite *Sarcoptes scabiei*³⁸, causes a highly contagious skin disease ..."

New text: "In contrast to mostly non-pathogenic and single-host *Demodex*, the multi-host mite *Sarcoptes scabiei*, also belonging to an ancient lineage showing codivergence since the time of the marsupial/placental dichotomy⁴⁰, causes a highly contagious skin disease."

R1.Q5b. L 158 please add unit to 140 (mm/year?).

R1.A5b. In the text, we added the units "mm/month" where applicable and a clarification at the beginning of the relevant section (L150):

Old text: "The variable *Average Precipitation* was fit ..."

New text: "The *Average Precipitation* variable (mean monthly precipitation, mm) was fitted ..."

R1.Q5c,d. L 158 eliminate ", which". d) L 158 eliminate "and"

R1.A5c,d. To account for the reviewer's suggestions, we rewrote this sentence for clarity.

Old text: The variable *Average Precipitation* was fit by nearly a bell-shaped curve with an upturn (significant, dense data), a peak at about 140 mm/month followed by a downturn, which in the area of scarce data and not statistically significant (Fig. 3c).

New Text: "The *Average Precipitation* variable (mean monthly precipitation, mm) was fitted as a shallow, concave downward arch with a peak at 173 mm/month (Extended Data Fig. 1 k)."

R1.Q5e.L174 it IS STILL POSSIBLE

R1.A5e. Changed as requested "it still likely"-> "it is still possible"

R1.Q5f. L 245 eliminate "a" after lacking

R1.A5f. This comment is not applicable anymore as the relevant portion of the manuscript has been re-written following the general comment of reviewer R4.

R1.Q5g. L 251 "while only a few predictors were related to hosts". How about factor PredictedMitesShare, the most impactful factor. It is host related (L 356). It should be mentioned here.

R1.A5g. This comment is not applicable anymore as the relevant portion of the manuscript has been re-written following the general comment of reviewer R4.

R1.Q5h. L 276 eliminate "two"

R1.A5h. See also R3.A2. This was an inadvertent mistake. We corrected this to reflect the actual number of risk-group species of *Notoedres* (five).

R1.Q5i. L 276 how about "the genus *Notoedres* originated as bat parasites"

R1.A5i. Changed as requested. A monophyletic lineage is assumed to have a single ancestor, hence we used singular.

Old text: "The genus *Notoedres* is originated from a bat parasitic ancestor"

New text: "The genus *Notoedres* originated as a bat parasite, ...".

R1.Q6. Miscellaneous comments. Successful host switches, leading to establishment on a new host species) are probably not very common in natural situations. Even switches to new host individuals of the same species may require immune-compromised hosts (infections of healthy hosts may be self-limiting in e.g. *Sarcoptes*). As also noted by the authors, most switches end in the death of the parasite. I do wonder if there would be worthwhile to try to model overall switching probability vs. successful switches (no idea how to do that).

R1.A6. Our analyses model successful host switches, i.e., each host record represents a parasite capable of forming sustainable populations on the host and being transmitted to new host individuals. Modeling the overall switching probability is an interesting topic, but it is outside the scope of our study.

All of the above is largely nit-picking and most likely will not influence the results of the analysis. Overall this is a very worthwhile study.

Reviewer #2 (Remarks to the Author):

I would like to congratulate the authors on conducting a comprehensive and insightful study on a topic of major importance. The assembly of such a large and complete dataset to analyze the factors affecting the transition of single-host parasites to multi-hosts is indeed a commendable effort. The study provides a robust quantitative framework that can potentially guide future research in this area. I have no major comments; here are just a few comments and suggestions that might help in further refining the manuscript:

Abstract:

R2.Q1. Line 17-19: To provide a clearer context right from the beginning, I would suggest a minor edit in the phrase "When single-host mites are". It could be revised to "When parasitic single-host mites are" to emphasize the parasitic nature of the mites being discussed.

R2.A1. Changed as requested.

R2.Q2. Line 27-30: In the section where the forecasting of Chiroptera (bats) and Carnivora as hosts with a high number of parasites in the multi-host risk group category is mentioned, it might be beneficial to elaborate slightly on why this result is significant and how it integrates with the broader findings of the study. Line 29-30: The phrase "that are probably capable of causing an

epidemic". It might be more informative to specify the kind of epidemic that is being referred to here, to give readers a clearer picture of the potential implications.

R2.A2. We modified this text as follows. Particularly, we clarified that *Notoedres* are bat skin-parasites.

Old text: When our model was used for forecasting, it identified Chiroptera (bats) and Carnivora as hosts having the largest number of parasites belonging to the multi-host risk group category. Of them, several single-host bat parasitic species of *Notoedres* were identified as having the potential to become multi-hosts that are probably capable of causing an epidemic.

New text: Our model predictions revealed an overrepresentation of mites associated with Rodentia (rodents), Chiroptera (bats), and Carnivora in the multi-host risk group. This highlights both the potential vulnerability of these hosts to parasitic infestations and the risk of serving as reservoirs of parasites for new hosts. In addition, we found independent macroevolutionary evidence that supports the prediction of several single-host species of *Notoedres*, the bat skin parasites, to be in the multi-host risk group, demonstrating the forecasting potential of our model.

Main Text:

R2.Q3. Line 39-61: Given that host specificity is central to this study, incorporating a discussion or mentioning the widely recognized three facets of host specificity (raw, phylogenetic, and geographic), as considered in the study by Poulin, R., Krasnov, B. R., & Mouillot, D. (2011), might be beneficial.

R2.A3. The primary concept introduced by Poulin et al. (2011) suggested integrating host specificity with either host phylogenetic diversity or geographic information. While these metrics may be valuable in certain contexts, applying them to the global mammalian dataset could be misleading. For instance, the average mammal phylogenetic diversity tends to be higher in tropical regions compared to temperate ones (Rosauer et al., 2015). Thus, incorporating phylogenetic diversity as a measure of host specificity would yield incompatible metrics across these diverse geographic areas. In contrast, our approach involves modeling the host ranges of parasites using both host phylogenetic diversity and geographical overlap (along with other variables) as independent predictors, aligning with the standard methodology in global predictive modeling (Albery et al., 2020). With all respect to the reviewer, we prefer to treat host phylogenetic distances and geography as independent variables, and not to use the metrics proposed by Poulin et al. (2011) in our study.

Albery, G. F., Eskew, E. A., Ross, N. & Olival, K. J. Predicting the global mammalian viral sharing network using phylogeography. *Nature Communications* 11, doi:10.1038/s41467-020-16153-4 (2020).

Poulin R, Krasnov BR, and Mouillot D. 2011. *Trends in Parasitology* 27 (8): 355-361.

Rosauer DF, and Jetz W. 2015. Phylogenetic endemism in terrestrial mammals. *Global Ecology and Biogeography* 24 (2): 168-179.

Material and Methods:

R2.Q4. Line 495: It would be great to see a bit more detail regarding the rationale behind the formula used in this section.

R2.A4. The original paragraph is "The latter model weights records by Google Scholar publication counts for each mite species as follows: $0.2 + (\text{totalPubs} \geq 10) * 0.8$."

We incorporated the following explanation of the rationale behind the formula: "This model accounts for the fact that some mite species received more research efforts than others, and thus their records of association with hosts are more complete than other species. This formula

gives full weights to mite species with more than 10 publications on Google Scholar and 0.2 weights to the species that have less than 10 publications. This weighting scheme was designed to mitigate the impact of unobserved multi-hosts, which are likely to be associated with a lower sampling effort, hence having lower publication counts.”

Figures:

R2.Q5. Figure 2b: To enhance the visual representation, it might be beneficial to order the data from higher to lower values, facilitating a more intuitive understanding of the data distribution.

R2.A5. This plot (Fig 2b) displays mean and confidence intervals of exponentiated model coefficients (odds ratios). The higher values (dots) indicate a positive effect on the dependent variable, while lower values suggest a negative effect, although this may not necessarily represent statistical significance. When error bars cross 1 (indicating no change in odds), it implies that the estimated effect of the predictor variable is not distinguishable from no effect. Therefore, despite their high absolute magnitudes, such variables may not be considered meaningful contributors to the model. In other words, (i) ordering the model coefficients by their magnitudes may lead to the misconception that high but statistically insignificant coefficients are meaningful for the model. Additionally, (ii) altering the order of variables could disrupt the continuity of the spline variables, which are particularly important as they illustrate changing trends within a single predictor. Furthermore, (iii) across all sections including Results, Discussions, and Methods, as well as in Tables, the independent variables are organized according to their major categories (mite-related, host-related, habitat disturbance). Thus, to ensure coherence in our data presentation, we believe it is best to preserve this order in Fig 2b.

Reviewer #3 (Remarks to the Author):

This manuscript models the probability of single-host (or specialist) mites becoming multi-host (or generalist) parasites. The study uses an impressive dataset of mite-host associations and accounts for many variables that could potentially play a role in the shift from single-host to multi-host. The details of the modeling approach are outside my expertise, but overall I think the study is thorough and the conclusions are sound based on the methods and the results. I found the figures to be informative and easy to interpret. I think this research is a valuable contribution to understanding host-parasite interactions and can inform the associations formed by other parasites. I have no major critiques, but do have some questions and feedback noted below.

Main Section

R3.Q1. I think it would be helpful to include a little more information about the mite, host, and abiotic traits that can influence if a mite is single-host or multi-host. It doesn't need to be a thorough explanation, but a bit of background would be helpful to understand the variables as they are explained in the "Factors affecting the odds to become multi-host" section.

R3.A1. We added the following explanation: "... We populate this dataset with a set of predictor variables related to parasites, their hosts, and the environment, which can aid in predicting mites' host range expansion. For example, among these predictors, two variables are related to mites' feeding specialization — mites with direct immune system interactions (e.g., hair follicular mites) are expected to have a lower establishment probability (and therefore narrower host range) compared to those feeding on non-immunogenic host tissue derivatives (e.g., fur mites). Similarly, ectoparasitic mites with diverse dispersal stages and broad geographic distributions are expected to have higher probabilities for successful transmission to new hosts, potentially contributing to broader host ranges. Hosts with many closely related

mammal species provide more opportunities for their parasitic mites to invade additional host species due to similar immune evasion mechanisms²⁷. Likewise, hosts living in regions with a high concentration of sympatric mammal species present direct opportunities for host shifting of their parasites, leading to broader host ranges. Other host properties, such as litter size, domestication status, and living in anthropogenically disturbed areas also offer further avenues for mite transfers and host range expansions. Finally, abiotic factors, such as temperature and humidity, may affect mite survival outside the host during transmission, thereby facilitating or prohibiting the transmission process, leading to broader or narrower host ranges, respectively."

Discussion Section

R3.Q2. Lines 271–276: there is a list of four species of sarcoptid skin mites, the following sentence refers to "these two single-host parasites" and it isn't clear what two that statement is referring to.

R3.A2. This was an inadvertent mistake. We changed it. See also R1.A5h.

R3.Q3. Lines 279–280: "In other words, in the past, inter-ordinal . . ." something about the wording of that sentence is not correct. I think it may just be missing "by" between "followed" and "a", but I'm not certain.

R3.A3. The reviewer's suggestion is correct. We inserted "by" between "followed" and "a". Now the sentence reads: "In other words, in the past, inter-ordinal host shifts followed by a speciation event occurred on a macroevolutionary scale in this genus. "

Methods Section

R3.Q4. Line 350: I think that it should be "host-parasite association" instead of "host parasite-association"

R3.A4. The reviewer's suggestion is correct. We changed "host parasite-association" to "host-parasite association"

R3.Q5. Lines 394–401: This section explains the biogeographic regions and the different coding schemes, but it does not explain why the last scheme (singleregion and multiregion) were used in the analyses. Why was that scheme used?

R3.A6. We added the following clarification: "We used the 2-category scheme in our final analyses because the 10- or 6-category schemes did not significantly improve the group separation (single- vs. multi-host mites) in the dependent variable."

Reviewer #4 (Remarks to the Author):

Review for Klimov and He

This paper addresses an interesting question in assessing the predictors of multi-host capacity in mites, with the added contribution of assembling a large dataset of host-mite associations. It's a valiant attempt at these complex analyses with some good aspects to the paper, but altogether it comes across as relatively hastily conducted and with many gaps in construction, interpretation, and presentation. As such, I can't recommend its publication. Apologies for not being more positive. Some of my more major comments are below.

On the positive side:

- The database is a very useful contribution to the literature on host-parasite associations.

- The models are interesting and advanced.
- The figures are aesthetically pleasing and apparently extremely well made and indicative of strong analytical expertise.

We systematically addressed the reviewer's concerns regarding model construction, result interpretation, and presentation. Please see below for details..

Misc:

R4.Q1. • The title is far too general, and should absolutely not be allowed to be published as it is. This is a paper about mites, and the title should be restricted to reflect that. The title gives the impression that it is a paper about parasites in general, and a review of drivers rather than an analysis, and gives no indication of what was actually found.

R4.A1. We accepted the suggestion and revised our title to: "When do single-host parasites become multi-host? Predictive host range modeling using a global mammalian-acarine dataset"

R4.Q2. • The structure of the discussion and results are difficult to tie together when reading, and the changeable and imprecise language used makes it difficult to link them to each result in the abstract.

R4.A2. We re-wrote these three sections to account for the reviewer comment. We believe that our revised manuscript has a clear structure in results and discussion, using consistent and precise terminology.

Approach:

R4.Q3. • The observation that a given mite has only been seen on members of one species does not necessarily make it a "single-host" parasite, as there may just be unobserved hosts, as the authors themselves acknowledge (line 210-213). At best, the analysis is picking up a reduced mean host range that is reflected more often at the species level as an observation of only one known host in a number of species. The number (and proportion) of actual "single-host" specialists is completely unknown, as far as I am aware. I am very doubtful that a subsampling approach like the one described could manage to overcome this data limitation. Feeding data into a reclassification model based on that same dataset and then restricting it and analysing a subset provides (in my opinion) false hope that these data hurdles have been overcome.

R4.A3. First, we would like to clarify that we separated our dataset into training (80%) and testing (20%) datasets; and the test dataset has **never** entered the model building stage. The down-sampling step the reviewer referred to was applied to solve the unbalanced sampling in the training dataset. We did not feed the same data in the model testing stage and always used different datasets to build a model and evaluate its performance.

Second, to comprehensively address the comment on single-host parasites being unobserved multi-host parasites we added a Positive and Unlabeled (PU) Learning analysis. This method explicitly assumes that the positive class (i.e., multi-host) is accurately labeled, while the negative class (i.e., single-host) represents an unlabeled mixture of negative and positive instances (single-hosts and unobserved multi-hosts, respectively). The results of this analysis closely mirrored those obtained from our other approaches, including down-sampling (mentioned by the reviewer) and weighting by past research efforts (not explicitly mentioned by the reviewer). For changes, see the new manuscript text directly.

While our PU learning analysis directly addressed the reviewer suggestion, we believe it's still important to provide a detailed response regarding our down-sampling methodology to

prevent any misunderstandings (i.e., comment "Feeding data ... and analysing a subset ..."). Down-sampling can be used to reduce noise in the majority class in situations where there is a class imbalance. In such cases, the majority class (i.e, single-hosts+unobserved multi-hosts) often dominates the dataset, potentially leading to overfitting or biased model performance. Down-sampling involves randomly removing instances from the majority class until the dataset is more balanced between classes. This can help prevent the model from being overly influenced by the majority class and improve its ability to accurately classify instances from both classes. In our case, a down-sampling strategy removed 60.3% (442 out of 733) of data in the single-host class. As a large portion of true single-hosts and potential unobserved multi-hosts have been removed, the overall influence of the multi-host class was effectively maximized when building our model. Furthermore, to evaluate the performance of our model, we used an independent test (holdout) dataset, and this dataset was not modified in any way (i. e. it was class-imbalanced). Therefore, in terms of predictive power of the multi-host class, our model should realistically reflect what one would see in 'the real world'.

In summary, here we added a comprehensive PU learning analysis, which directly addresses the aspects related to unobserved multi-hosts. We also make it clear in the text that we build and test our models using different datasets, and added the following phrase to emphasize that our sample weighting by research efforts was also done as part of our broader efforts to reduce the effect of unobserved multi-hosts: "The weighted model weights records by Google Scholar publication counts for each mite species as follows: $0.2 + (\text{totalPubs} \geq 10) * 0.8$. This model accounts for the fact that some mite species received more research efforts than others, and thus their records of association with hosts are more complete than other species. This formula gives full weights to mite species with more than 10 publications on Google Scholar and 0.2 weights to the species that have less than 10 publications. This weighting scheme was designed to mitigate the impact of unobserved multi-hosts, which are likely to be associated with a lower sampling effort, hence having lower publication counts".

R4.Q4. • If the analysis was phrased as “factors contributing to host range in mites” rather than “switching from one to multi host” that would solve a lot of the approach-related problems the paper suffers from, and the models could stay much the same as long as the interpretation was modified in the writing and appropriate attention was given to sampling bias. I’m confused why the authors approached it in this binary way rather than simply proceeding with the host count data they have.

R4.A4. Both of these statements are correct and complementary. The former statement, 'factors contributing to host range in mites' (1), emphasizes the dependent variables, or predictors, while the latter, 'switching from one to multi-host' (2), emphasizes the independent variable, that is, what we are predicting. Our dependent variable in statement 2 refers to the outcome being predicted, with two levels: single-host and multi-host, and we predict them based on a set of independent variables (statement 1). This general approach is standard, well-understood, and widely used. We believe that predicting single-host versus multi-host is a biologically interesting question, which can be modeled using a robust statistical framework, while accounting for various biases, such as class imbalance and unobserved multi-hosts. It also makes it easy to analyze non-linear predictors. However, Reviewer 4 identifies 'problems' associated with this approach without specifying them in the review and proposes predicting 'host counts' instead.

Following this suggestion of reviewer 4, we conducted a linear regression analysis to predict host counts (see results below). Although this analysis yielded similar results in terms of significance and directionality for many predictors, the dependent variable (host counts) exhibited non-normal distribution, as evidenced by clear trends and departure from the expected distribution in both residual (Fig. a) and quantile plots (Fig. b). The violation of the normality

assumption in the dependent variable undermines the accuracy and reliability of regression coefficients, potentially leading to misleading conclusions. For example, significance in the variable HostDomestication may be erroneously inferred due to a limited number of clustered data points, as illustrated in Fig. c.

Coefficients:

	Estimate	Std. Error	t value	Pr(> t)
(Intercept)	0.6204793	0.4721911	1.314	0.189131
ImmuneResponse2SECONDARY	1.0785919	0.1999400	5.395	8.56e-08 ***
ImmuneResponse3NONE	0.2199774	0.1276049	1.724	0.085034 .
Parasitism2ECTO	0.5471604	0.1441952	3.795	0.000157 ***
BioRegCoding3Multiregion	1.8547405	0.2257420	8.216	6.40e-16 ***
MiteDispersalStageF	-0.1124589	0.4083514	-0.275	0.783068
MiteDispersalStageF_IMM	0.0684774	0.3774979	0.181	0.856092
PrecopulatorGuardingN	-0.0399859	0.1524055	-0.262	0.793093
max_JC50	0.0201008	0.0033843	5.939	3.93e-09 ***
max_PD10	0.2769906	0.0281307	9.847	< 2e-16 ***
avg_HostLitterSize	0.0111482	0.0289095	0.386	0.699856
hostDomY	2.0287167	0.3788811	5.354	1.06e-07 ***
avg_HostBodyMass_g_log	-0.0493978	0.0257702	-1.917	0.055538 .
avg_prec	-0.0020043	0.0011145	-1.798	0.072406 .
avg_temp	-0.0007539	0.0008411	-0.896	0.370317
avg_HumanPopDen_5p_n_per_km2_log	0.2261096	0.0518233	4.363	1.41e-05 ***

Signif. codes: 0 '***' 0.001 '**' 0.01 '*' 0.05 '.' 0.1 ' ' 1

Residual standard error: 1.517 on 1008 degrees of freedom

Multiple R-squared: 0.3416, Adjusted R-squared: 0.3318
F-statistic: 34.86 on 15 and 1008 DF, p-value: < 2.2e-16

Residuals:

Min	1Q	Median	3Q	Max
-4.9594	-0.5367	-0.1620	0.2654	20.3470

R4.Q5. • The predictors are relatively underpresented in the introduction in terms of hypotheses for why specific mite and host traits should create variation in host range.

R4.A5. We updated the Introduction section following the reviewer comment, for detail, see R3.A1.

Model:

R4.Q6. • Combining two variables (phylogenetic distance and geographic overlap) into one explanatory variable is an odd and unjustified approach that could do extremely strange things that would be avoided entirely by including both variables independently. I think this was carried out because the analysis as designed required a single variable? But why could these variables not be included in the analyses as they were?

R4.A6. We conducted a new analysis where both geographical overlap and phylogenetic distances were included as separate variables. For details and explanations see R1.A3.

R4.Q7. • The analysis is a mite-level analysis I think (the unit of replication is the mite, where each species is a unique value. But a lot of these variables are related to the host, and it's unclear how each value was reached. Some variables it's explicitly max, some it's mean, and others give no info. Or was the unit of replication a mite-host association, such that each host was represented multiple times, and I've misunderstood? More clarity needed.

R4.A7. To account for the reviewer comment, in M&M, we (i) explicitly specified whether the mean or max was taken for each relevant variable and (ii) added the following explanations: "Our analysis is a mite species-level analysis, i.e., each mite species (or subspecies) has a set of values related to the mite itself, host(s), or the environment. In the case of multi-host mites, we averaged the values of corresponding host-related predictor variables to ensure that the combined effect of all hosts is appropriately represented. For two variables, *Co-Distributed Potential Hosts* and *Phylogenetically Similar Potential Hosts*, we used maximum values because the effect of these variables is expected to be most pronounced at maximum values, which will be explained in the following section."

R4.Q8. • The sharing model itself is very interesting and I'm not sure why it was so sidelined and reduced to producing an explanatory variable in another model.

R4.A8. This is a great suggestion. We have expanded our discussion on the mites sharing network and its implications. We no longer use it to produce the explanatory variable, as phylogenetic distance and geographic overlap are now treated as separate variables in the mites' host range model.

Writing:

R4.Q9. • The paper is fairly inaccurately written throughout: for example, on line 25, what is "the proximity to the host immune system"?

R4.A9. We re-wrote this phrase for clarity as follows:

Old text: Among mite-related variables, the most important was the proximity to the host immune system which was correlated with the mouthpart morphology.

New text: “This model identified statistically significant predictors related to parasites, hosts, climate, and habitat disturbance. The most important predictors included the parasite’s contact level with the host immune system and two variables describing host phylogenetic similarity and spatial co-distribution. ”.

R4.Q10. • The discussion is roundabout and poorly tied to the authors’ results.

R4.A10. There are no specific details here. To address this comment, we extensively revised the Discussion section. See the manuscript text directly.

R4.Q11. • Related, the grammar needs checking thoroughly. E.g. line 75 “acariform mites mite-mammal-associations” is fairly sloppily written. Line 77, “all known target parasites and their all known hosts”, Etc.

R4.A11. These phrases were corrected to "acariform mite-mammal associations" and "all known target parasites and all their known hosts are included", respectively. Being non-native speakers, we do apologize for these mistakes. We asked a native speaker to review the manuscript and enhance its English.

R4.Q12. • Beyond this, it is also excessively jargony in points – the paragraph beginning line 133 has a lot of GAMM-specific wording that would be very difficult to dissect for an unfamiliar reader particularly if they haven’t yet read the methods.

R4.A12. This comment is not relevant anymore because it applies to the old variable, *PredictedMitesShare*. We re-coded this variable into two separate variables (see R1.A3 and R4.Q6). However, in the text, we explain our natural spline regression methodology (GAMM is a related technique) for a different variable as follows: “Briefly, a natural spline function, is a flexible curve-fitting technique that smooths the data without imposing strict assumptions about the relationship between the predictor variable and the outcome variable. This approach is especially useful when the relationship is nonlinear or when the effect of the predictor can change directionality. For example, in scenarios where midrange temperatures are optimal and have a positive effect on the outcome, while both low and high temperatures diminish this effect, a natural spline can effectively capture this nonlinear relationship.”

R4.Q13. • The analysis is very complex, and the “results first” format is doing it no favours. Lots of variables and results come out of the blue with no context, making it very difficult to follow the thread of importance.

R4.A13. This comment is similar to query R4.Q5 above. To address the reviewer's comment to add more context, we described our predictor variables in the Introduction section. See R3.Q1 for a detailed explanation. The 'results first' rule is a requirement of the journal, and we are unable to deviate from it.

R4.Q14. • The reader is left with relatively little idea of what to take from the paper on finishing the discussion; this analysis is doing a lot of different things, and none of them is given sufficient time or attention, such that nothing in particular shines through.

R4.A14. We created a predictive model that can be useful for forecasting epidemic risk-group parasites. We re-wrote the Discussion section to emphasize this. Particularly, we modified the conclusion paragraph of the Discussion to make it clear that this predictive model is the most important outcome of our work (see below):

“In conclusion, we assembled the largest and the most complete dataset to date on mites permanently parasitic on mammals and developed a predictive model to analyze a set of determinants influencing the likelihood of single-host parasites transitioning into multi-host

parasites. Our model accounted for potentially unobserved host-parasite links and class imbalances, identifying statistically significant predictors related to parasites, hosts, climate, and habitat disturbance. This analysis provided valuable insights into the ecological and epidemiological aspects of mammalian acarine parasites and potential disease transmission dynamics. When applied to forecast epidemic risk-group parasites, our model revealed that rodents (Rodentia), bats (Chiroptera) and Carnivora harbor a disproportionately large number of single-host parasites with the potential to become multi-host, including the sarcoptid skin mites of the genus *Notoedres*, posing significant epidemic risks. Our study is the first attempt to analyze host specificity patterns in mammalian acarine parasites in a predictive, quantitative framework. However, more empirical and experimental studies are clearly needed to understand the general properties underpinning host-parasite interactions.”

R4.Q15. • 68-70: I'm not sure this assertion is either true or addressed by this paper.

R4.A15. We re-wrote the relevant text to make it clearer that these aspects will be addressed in our paper.

Old text: "However, despite progress extolled by these advanced modeling approaches, it is still unclear whether host range predictions can be improved by using a global, pan-host, and pan-pathogen context, and whether their general conclusions are held in other host-pathogen systems.

New text: "Despite progress extolled by these advanced modeling approaches, it still remains unclear whether host range predictions drawn from these studies are applicable to other host-pathogen systems.“

R4.Q16. • 150-153: Why interpret a non-significant result? This interpretation should be in the discussion, anyway, not the results. Same with the marginal result at 170-172.

R4.A16. We removed the elements of discussion or interpretation.

R4.Q17. • 248-250: I'm no closer to understanding this variable (which seems to combine two very different traits) than I was in the abstract. Having read the methods I now get it, but it needs far more context adding where it's mentioned before the methods.

R4.A17. This suggestion is not relevant anymore because it applies to the old variable, *PredictedMitesShare*. We re-coded this variable into two separate variables, see R1.A3 and R4.Q6.

REVIEWERS' COMMENTS

Reviewer #1 (Remarks to the Author):

The response to initial reviewer comments is appropriate. All my earlier issues have been addressed in satisfactory fashion and I have no further significant problems with this study. It should be published.

I found two minor issues with phrasing that can be addressed:

L115. “..whether a single-host parasite can extend its host range to become multi-host.” Correct phrasing? Authors seem to be working on probability that such a thing occurs, so perhaps “..the likelihood that a single-host parasite can extend its host range to become multi-host.”

L249 eliminate “be” to get to “.. increases to over 0.75 ..”

Reviewer #1 (Remarks on code availability):

I do not have the knowledge to evaluate codes, so it makes little sense to examine it in detail.

Reviewer #3 (Remarks to the Author):

In a previous review of this manuscript I noted that it is a valuable contribution to our understanding of host-specificity and develops a method for researching how parasites might shift from a single host species to multiple host species. My previous comments were relatively minor and they have been thoroughly addressed. I appreciate the work the authors put into revising the manuscript and addressing the reviewers' concerns.

Reviewer #4 (Remarks to the Author):

The authors have done a commendable job of responding extensively and in detail to quite a hefty set of reviews. Their manuscript is much improved as a result; I recommend accepting it as is.

Response to Referees

Our point-by-point responses to reviewers' comments are provided below.

REVIEWER COMMENTS

Reviewer #1 (Remarks to the Author):

The response to initial reviewer comments is appropriate. All my earlier issues have been addressed in satisfactory fashion and I have no further significant problems with this study. It should be published.

I found two minor issues with phrasing that can be addressed:

L115. “..whether a single-host parasite can extend its host range to become multi-host.” Correct phrasing? Authors seem to be working on probability that such a thing occurs, so perhaps “..the likelihood that a single-host parasite can extend its host range to become multi-host.”

L249 eliminate “be” to get to “.. increases to over 0.75 ..”

R: L115, we modified the sentence to “we build a statistical model to predict *the likelihood* that a single-host parasite can extend its host range to become multi-host”
L249, we modified the sentence to “The sharing probability increases to over 0.75 when the host pair is fully sympatric, i.e., geographic overlap = 1.”